# Aspirin-responsive gene switch regulating therapeutic protein expression

Jinbo Huang [1], Ana Palma Teixeira[1], Ting Gao[2], Shuai Xue[1,2], Mingqi Xie [2,3] & Martin Fussenegger [1,4] ✉

Current small-molecule-regulated synthetic gene switches face clinical limitations such as cytotoxicity, long-term side-effects and metabolic disturbances. Here, we describe an advanced synthetic platform inducible by risk-free input medication (ASPIRIN), which is activated by acetylsalicylic acid (ASA/aspirin), a multifunctional drug with pain-relieving, anti-inflammatory, and cardiovascular benefits. To construct ASPIRIN, we repurpose plant salicylic acid receptors NPR1 and NPR4. Through domain truncations and high-throughput mutant library screening, we enhance their ASA sensitivity. Optimized NPR1 fused with a membrane-tethering myristoylation signal (Myr-NPR1) forms a complex with NPR4, which is fused with a DNA binding domain (VanR) and a transactivation domain (VP16). ASA induces dissociation of the Myr-NPR1/NPR4-VanR-VP16 complex, allowing nuclear translocation of NPR4-VanR-VP16 to activate VanR-operator-controlled gene expression. In male diabetic mice implanted with microencapsulated ASPIRIN-engineered cells, ASA regulates insulin expression, restores normoglycemia, alleviates pain and reduces biomarkers of diabetic neuropathy and inflammation. We envision this system will pave the way for aspirin-based combination gene therapies.

The pursuit of gene switches that are safe, precise, sustainable, and versatile, enabling the regulation of therapeutic human cells for personalized medicine, has been a longstanding goal in the field of synthetic biology[1–6]. Despite decades of research, however, inducible gene circuits devoid of side effects and safety concerns have remained elusive[7,8]. The classical Tet-On/Tet-Off systems that control gene expression via tetracycline analogs remain the most widely adopted approaches for mammalian gene regulation at present[4,9]. Nevertheless, their reliance on antibiotics is problematic, given the increasing issue of antibiotic resistance in clinical settings[9–11]. Therefore, scientists have explored alternative small molecules to regulate transgene switches, including protease inhibitors[12], food-related compounds[13,14], plant-derived molecules[15], cosmetic compounds[16], unnatural amino acids[17], peptides[18], and synthetic compounds (e.g., rapamycin)[19]. Despite these developments, small molecules so far reported to drive gene switches face a variety of limitations, such as cytotoxicity, off-target effects, development of resistance, metabolic disturbances,

long-term side-effects, complicated pharmacodynamics and low bioavailability[7–11,20–23], hampering their broad clinical application. Thus, there is a pressing need for a gene switch controlled by a safe, widely used, small-molecule medication for therapeutic protein expression in personalized medicine[1–3,5].

In this context, aspirin, a globally recognized medication with a rich medical history[24,25], is a compelling candidate. The medical use of willow bark extracts to treat fever, cold and pain dates back thousands of years[24,26]. Then in the 19th century, scientists isolated and synthesized the active compound in willow bark, salicylic acid (SA)[24–26]. Its derivative, acetylsalicylic acid (ASA), was later developed as a more palatable and stable derivative, which was mass-produced under the name aspirin, quickly becoming one of the most commonly used and affordable medications in modern society[24–26]. Its multifaceted therapeutic profile encompasses analgesic, antipyretic, anti-inflammatory, and antiplatelet properties, making it a go-to medication for many

[1]Department of Biosystems Science and Engineering, ETH Zurich, Basel, Switzerland. [2]Westlake Laboratory of Life Sciences and Biomedicine, Hangzhou, Zhejiang, China. [3]Key Laboratory of Growth Regulation and Translational Research of Zhejiang Province, School of Medicine and School of Life Sciences, Westlake University, Hangzhou, Zhejiang, China. [4]Faculty of Science, University of Basel, Basel, Switzerland. ✉e-mail: fussenegger@bsse.ethz.ch

common ailments such as headaches, muscle aches, and arthritis[25,27]. Indeed, one of the most well-established benefits of ASA is its ability to reduce the risk of cardiovascular events, notably heart attacks and strokes, due to its antiplatelet activity[28]. Consequently, ASA is widely recommended for individuals at risk of cardiovascular disease, including those with a history of heart attack or stroke, as well as those grappling with high blood pressure, diabetes, or high cholesterol levels[24,26–28]. Furthermore, recent investigations have expanded our understanding of its potential. For example, low-dose ASA is prescribed to pregnant women at risk of preeclampsia[29]. ASA also shows promise in inhibiting tumor growth, suppressing metastasis, and enhancing the efficacy of chemotherapy and immunotherapy[30]. There is also evidence that it reduces colorectal cancer incidence[31]. In view of its unparalleled accessibility, safety profile, and multifaceted therapeutic potential, we considered that integration of ASA into gene switch platforms would represent a transformative advance in biomedical research.

Mechanistically, ASA inhibits the activity of cyclooxygenase enzymes, which convert arachidonic acid into prostaglandin[26,28,32] (Supplementary Fig. 1). This reaction reduces prostaglandin synthesis, thereby suppressing inflammatory responses, relieving pain, reducing fever and preventing blood clots[26,28,30]. In the liver, ASA is metabolized into SA, which in turn is further metabolized to water-soluble compounds that are easily excreted via the kidneys[33,34]. Interestingly, SA itself functions as a natural plant hormone, being involved in defense mechanisms against pathogens[35]. SA disrupts the interaction between nonexpressor of pathogenesis-related gene 1 (NPR1) and NPR4, thereby initiating downstream gene expression to counteract pathogen attacks[35–37]. The striking structural similarity between ASA and SA underscores the tantalizing prospect of engineering signaling pathways responsive to ASA by leveraging the plant-derived SA-responsive proteins, NPR1 and NPR4.

In this study, we develop an <u>a</u>dvanced <u>s</u>ynthetic <u>p</u>latform <u>i</u>nducible by <u>r</u>isk-free <u>in</u>put medication (ASPIRIN) based on NPR1 and NPR4 to regulate the expression of therapeutic proteins for cell-based combinational therapy in response to ASA. This system not only delivers an ASA-induced therapeutic protein, but also takes advantage of the intrinsic activities of ASA itself. To construct the ASA sensor, we employ protein engineering and directed evolution/screening to increase the ASA sensitivity of NPR1 and NPR4. Then, to build the ASPIRIN system, we introduce an N-terminal myristoylation signal peptide (Myr-)[38] into an optimized NPR1 mutant (Myr-NPR1), enabling the retention of NPR1 on the inner cell membrane, where it forms a complex with NPR4 C-terminally fused with a DNA binding domain (VanR) and a transactivation domain (VP16) (NPR4-VanR-VP16). Upon exposure to ASA or SA, the complex dissociates, allowing NPR4-VanR-VP16 to translocate to the nucleus and activate gene expression downstream of the VanR operators. The system shows reliable biocompatibility and specificity, making it particularly suited for gene and cell-based therapies compatible with an anti-nociceptive control regimen. For a proof-of-concept study, we implant microencapsulated engineered monoclonal human cells expressing insulin under the control of the ASPIRIN system in a mouse model of type 1 diabetes (T1D). As expected, induction with ASA restores normoglycemia in the diabetic mice. Furthermore, ASA significantly reduces inflammatory biomarkers associated with diabetic complications. This illustrates the potential of combining conventional small-molecule drugs with cell-based gene therapy for treating diabetes and its complications. We anticipate that this system will pave the way for the development of other aspirin-based combination gene therapies.

## Results

### Design and characterization of an ASA-induced gene switch
To construct an ASA-responsive transgene switch in mammalian cells, we initially explored several bacterial-derived regulators, including

SalR, BenR, XylS, and MarR[39–42]. Given the structural similarity between SA and ASA, with SA being the main metabolite generated by ASA metabolism within the human body[33], we tested both ASA and SA as inducers. Despite extensive engineering efforts, none of these regulators could induce substantial transgene expression in an ASA-dependent manner (Supplementary Fig. 2). Therefore, we shifted our focus to NPR1 and NPR4 proteins, which have been shown to dissociate in the presence of SA in the plant *Arabidopsis thaliana*[35,37], suggesting their potential as components of an ASA/SA-responsive synthetic gene switch. To establish an ON-type system triggering transgene expression in the presence of inducers, we tagged NPR1 with a plasma membrane-tethered myristylation signal (Myr-NPR1), while NPR4 was fused to a synthetic transcription factor (synTF) comprising a transactivation domain (e.g., the herpes simplex virus protein 16 (VP16), or VP64, four tandem copies derived from VP16) fused to either bacterial tetracycline-dependent TetR (NPR4-TetR-VP64) (Supplementary Fig. 3a) or vanillic-acid-dependent VanR (NPR4-VanR-VP16) (Fig. 1a, b). Under basal conditions, the two components interact at the plasma membrane. However, in the presence of ASA/SA, the interaction between NPR1 and NPR4 is disrupted, leading to the release and nuclear translocation of NPR4-synTF, where it can bind its recognition DNA sequences, thereby inducing expression of the gene of interest (GOI) (Supplementary Fig. 3a and Fig. 1a, b).

As previously shown, NPR1 and NPR4 are homologous proteins containing a salicylic acid-binding core (SBC) in their C-terminal regions (Fig. 1c, d). By means of domain analysis[43] and secondary structure prediction[44], we designed several SBC-containing truncation variants of *NPR1* (Fig. 1c) and *NPR4* (Fig. 1d) to delineate the hypersensitive domains. Following combinatorial testing of full-length NPR1 and NPR4 and their truncation variants, most of them produced significant but lower than 2-fold inductions in the presence of ASA and SA. Nevertheless, the T2 truncations (NPR1-T2 and NPR4-T2) exhibited up to 3.8-fold induction (Supplementary Fig. 3b–i and Supplementary Fig. 4) and were selected for further optimization.

Comparative analysis of three transactivation domains (VP16, VP64 and VPR, a tripartite activator containing VP64, P65, and Rta) using VanR as the DNA-binding domain (DBD) revealed that VP16 enabled the highest fold-induction of NanoLuciferase (NLuc) reporter expression (Fig. 1e). Subsequent screening of the mass ratios of the three components of the gene switch, NPR1 (pJH140, $P_{SV40}$-Myr-NPR1-T2-pA), NPR4 (pJH196, $P_{SV40}$-NPR4-T2-VanR-VP16-pA), and reporter (pJH2012, $O_{VanO2}$-$P_{hCMVmin}$-NLuc-pA) plasmids (Supplementary Data 1-2), identified the highest fold-induction in NLuc expression (12.7-fold by ASA) at a ratio of 4:1:4, respectively (Fig. 1f–h).

### Directed evolution of the gene switch via random mutagenesis
In order to enhance the responsiveness of the genetic circuit to ASA or SA, we implemented error-prone PCR-based random mutagenesis on NPR1-T2 and NPR4-T2 proteins sequentially over multiple rounds. The resulting libraries were subjected to high-throughput functional screening assays by transfecting HEK-293T cells in 96-well plates, followed by NLuc secretion assays after 24 h of incubation with or without ASA (Fig. 2a–h and Supplementary Fig. 5). Initially, HEK-293T cells were co-transfected with wild-type (WT) NPR1-T2 (pJH140) and a library of NPR4-T2-VanR-VP16 variants (Fig. 2a). The mutants with increased fold-induction from the primary screening (Supplementary Fig. 5a) were further validated in a secondary screen (Fig. 2b). The NPR4-T2 mutant showing the highest NLuc fold-induction (NPR4$_{mut1}$: NPR4$_{K476R/A484T/R493K}$, Fig. 2b) was selected for pairing with a NPR1-T2 mutant library for the next round of screening (Fig. 2c, d and Supplementary Fig. 5b). From this process, we identified four mutants outperforming the WT NPR1-T2. The top NPR1-T2 mutant (NPR1$_{mut3}$: NPR1$_{Q543R}$, Fig. 2d) was paired with a new mutant library generated from NPR4$_{mut1}$. After the third and fourth rounds of mutagenesis screening for NPR4 (Fig. 2e, f and Supplementary Fig. 5c) and NPR1

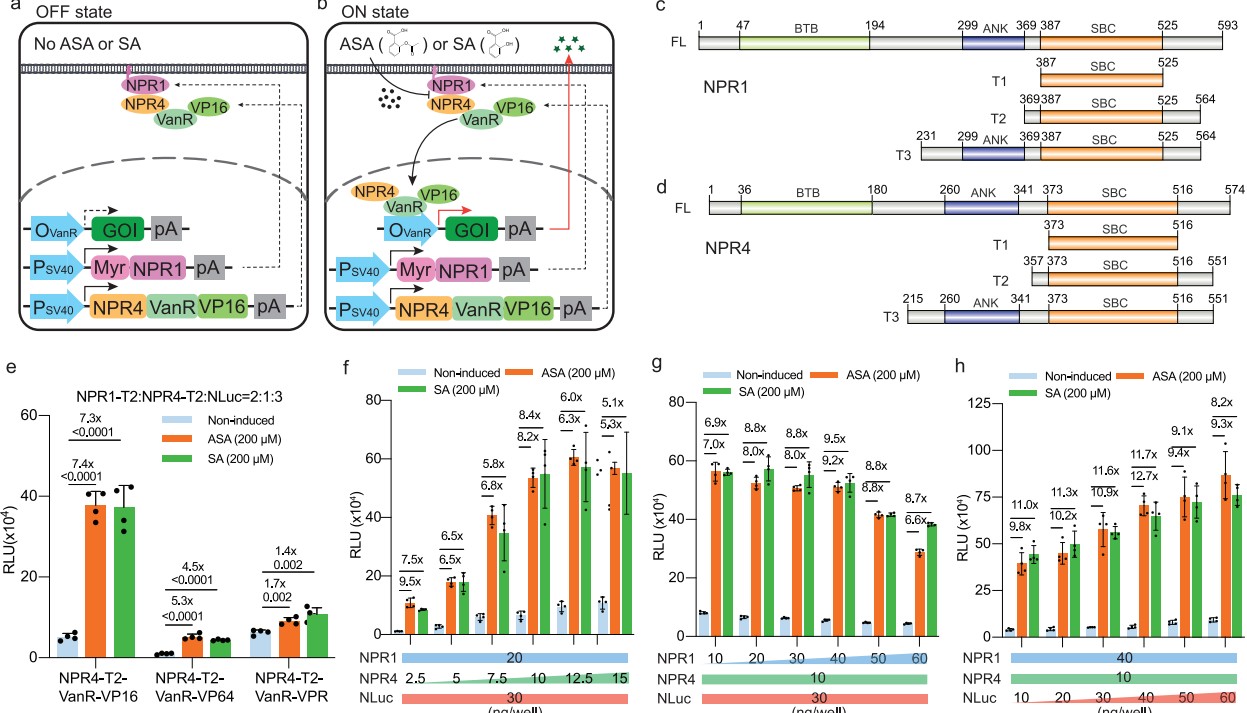

**Fig. 1 | Design of the ASA-activated transgene switch in mammalian cells.**
**a**, **b** Schematic illustration of ASA- or SA-regulated transgene switch. NPR1 tagged with a membrane-tethered myristoylation sequence (Myr-NPR1) and NPR4 fused to a synthetic transcription factor, consisting of VanR fused to the transactivation domain VP16, are constitutively expressed and interact at the plasma membrane in the absence of ASA or SA (**a**). Conversely, in the presence of inducers (**b**) the NPR1/NPR4 complex is disrupted, resulting in NPR4-VanR-VP16 translocation to the nucleus, where it binds to VanR-binding sites, activating expression of downstream gene-of-interest (GOI, green stars). **c**, **d** Domain analysis and truncation variants of NPR1 (**c**) and NPR4 (**d**). Both proteins contain an SA-binding core (SBC). BTB, broad-complex, tramtrack and bric a brac domain. ANK, ankyrin repeat domain. **e** NanoLuciferase (NLuc) reporter produced after 24 h by cells co-transfected with

NPR1, NPR4 and reporter genes. **f–h** Mass ratio optimization among NPR1 (pJH140), NPR4 (pJH196) and reporter plasmid (pJH2012). The system was transiently transfected with increasing amounts of one plasmid and constant amounts of the other two. The amounts of NPR1, NPR4 and reporter plasmids were applied per well of a 96-well plate as indicated. In **f–h**, all treatment groups showed significant differences compared to the control groups ($P$ values < 0.05). In **e–h**, all treatment groups were incubated with ASA/SA as indicated, while non-treated groups received an equivalent amount of vehicle (DMSO). Statistical significance was analysed by means of a two-sided unpaired t test. All data are means ± SD; $n = 4$. $P$ values were calculated versus the corresponding non-induced control. The induction factors are indicated. RLU: relative luminescence unit. Source data are provided as a Source Data file.

(Fig. 2g–h and Supplementary Fig. 5d), respectively, we identified a variant pair highly sensitive to ASA. The NPR4 variant containing five mutations (NPR4-T2mut6: NPR4-T2N391T/K476R/T480N/A484I/R493K) and the NPR1 variant containing three mutations (NPR1-T2mut6: NPR1-T2G391S/L427H/Q543R) significantly enhanced the NLuc fold-induction in response to ASA (from 12× to 44×). Interestingly, during sequential mutagenesis rounds, the alanine residue at position 484 (A484) of NPR4 was changed first to threonine and then to isoleucine (Fig. 2b, f). Furthermore, all the mutations of NPR4 are clustered within the SA binding pocket[35,36], suggesting the relevance of these positions to the ASA-inhibited interaction between NPR1 and NPR4. Additionally, point mutations of the evolved NPR4 residues to alanine or glycine completely abolished the system's ability to sense ASA and SA (Supplementary Fig. 6), highlighting the critical role of these evolved residues in ASA and SA sensing. To assess the localization of the Myr-NPR1-T2mut6 protein, we fused EGFP at its C-terminus (pJH236: PSV40-Myr-NPR1-T2mut6-EGFP-pA). Fluorescence microscopy analysis confirmed predominant retention of Myr-NPR1-T2mut6 in the plasma membrane, regardless the presence or absence of ASA or SA (Fig. 2i–n and Supplementary Fig. 7a–c). Additionally, co-transfection experiments involving NPR4-T2mut6-VanR-VP16 tagged with mCherry at its C-terminus (pJH237: PSV40-NPR4-T2mut6-VanR-VP16-mCherry-pA), confirmed that NPR4-T2mut6 was mainly localized in the plasma membrane in the uninduced state (Fig. 2i, j and Supplementary Fig. 7d–f). However, upon ASA or SA exposure, NPR4-T2mut6 was released from the membrane and translocated to the nucleus

(Fig. 2k–n). Moreover, co-immunoprecipitation (Co-IP) and western blotting (WB) analyses confirmed that ASA and SA indeed effectively mediated the association and dissociation of truncated NPR1-T2mut6 and NPR4-T2mut6 (Supplementary Fig. 8). These data collectively validate the functionality of the designed ASPIRIN system, utilizing ASA or SA as an inducer to regulate transgene expression in mammalian cells.

## Characterization and validation of ASPIRIN system

The dose-response curve demonstrated a nearly linear relationship between ASA or SA concentration and reporter expression in the 10 μM to 250 μM range, with EC50 values of 70.5 μM and 46.4 μM, respectively (Fig. 3a). Notably, tested ASA or SA showed no significant impact on cell viability at concentrations below 500 μM (Supplementary Fig. 9a, b). Assessing the versatility of the system in various mammalian cell lines, we observed differences in expression levels and fold-inductions following ASA or SA treatment, but the functionality of the system was confirmed across all tested cell lines (Fig. 3b and Supplementary Fig. 9c). Our results confirm the compatibility of the ASPIRIN system with biopharmaceutical manufacturing cell platforms (HEK-293T, CHO-K1, and BHK-21), colon (Caco-2) and liver (HepG2) tissues, and human mesenchymal stem cell-derived hMSC-TERT (Fig. 3b), which are gaining prominence as candidates for advanced cell-based therapies. These findings underscore the potential of the ASPIRIN system for diverse applications. On the basis of fold-induction and basal and maximal expression levels, HEK-293T cells were selected for subsequent experiments.

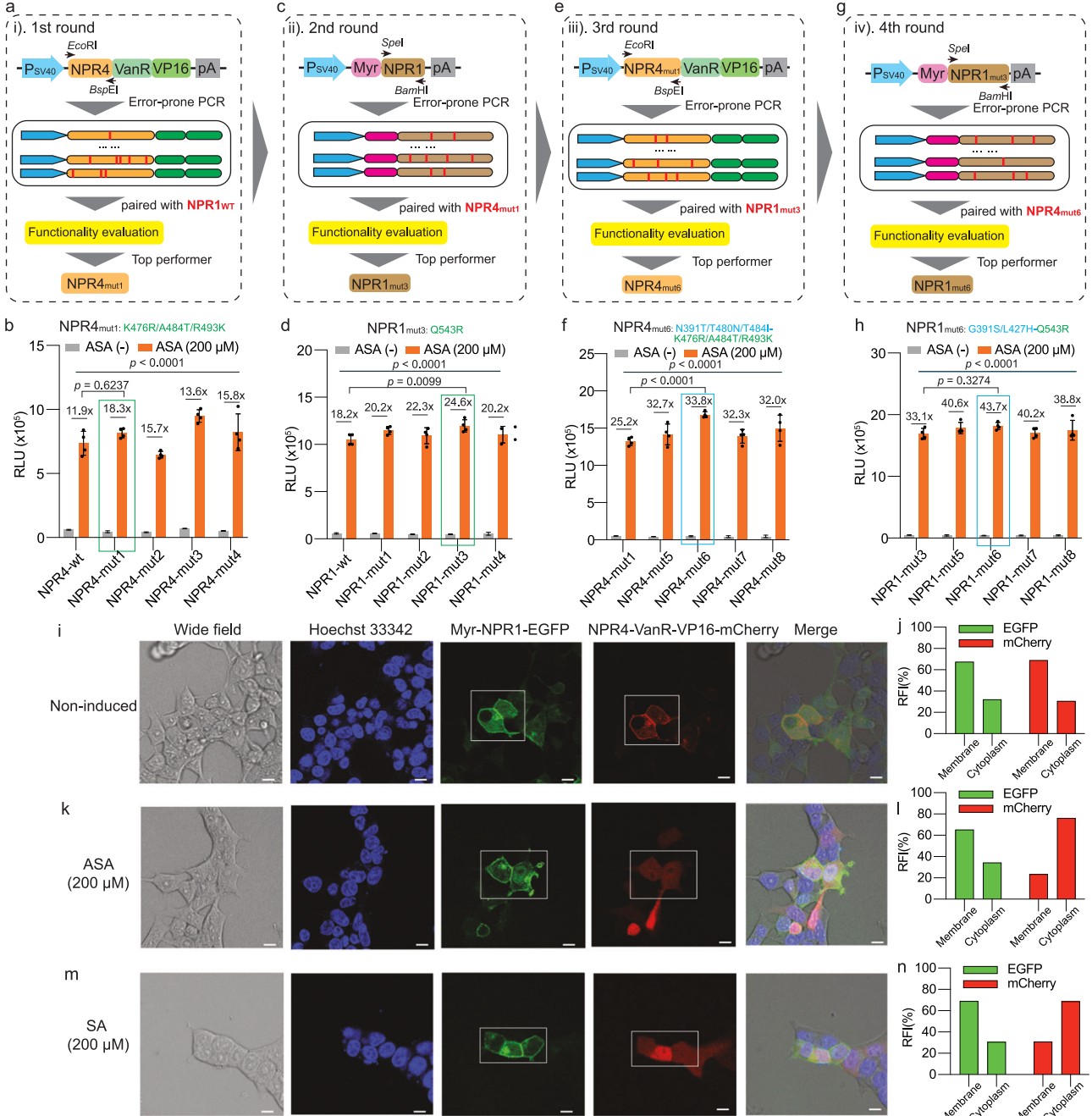

**Fig. 2 | Directed evolution of NPR1 and NPR4 by random mutagenesis. a** First round of mutagenesis for NPR4-T2 using error-prone PCR. WT NPR1-T2 was employed for functionality evaluation. **b** Confirmation of the top four NPR4-T2 mutant variants. The best-performing variant, NPR4mut1, was selected. **c** Second round of mutagenesis for NPR1-T2 paired with NPR4mut1. **d** Functionality evaluation for the top four NPR1-T2 mutant variants. The most effective variant, NPR1mut3, was chosen. **e** Third round of mutagenesis. NPR4mut1 was paired with NPR1mut3. **f** Functionality evaluation for the NPR4 mutant variants. The optimal variant, NPR4mut6, was selected for further analysis. **g** Fourth round of mutagenesis. NPR1mut3 was paired with NPR4mut6. **h** Functionality evaluation for the NPR1 mutant variants. The best-performing variant, NPR1mut6, was chosen for further experimentation. **i–n** Fluorescence microscopic imaging and fluorescence intensity

quantification of HEK-293T cells co-transfected with pJH236, pJH237 and pJH2012 in the absence (**i, j**) or presence of ASA (**k, l**) and SA (**m, n**). Fluorescence imaging of EGFP (488/510) and mCherry (587/610) was performed using a Leica SP8 confocal microscopic platform. Hoechst 33342 (380/420) was utilized for nuclear localization. **j, l, n** EGFP and mCherry intensity quantification from the squared regions of **i, k, m**, respectively. RFI: relative fluorescence intensity. A representative image from each group of three replicates is shown. Scale bar (white): 10 μm. In **b, d, f, h**, the induction factors are indicated, and all induction groups are significantly different from the indicated control groups ($P < 0.0001$). Statistical significance was analysed by means of two-sided two-way ANOVA with Tukey's test between groups and one-sided unpaired t test within groups. All data are presented as means ± SD; $n = 4$. Source data are provided as a Source Data file.

Moreover, NLuc secretion into the culture medium was proportional to the concentration of ASA (Fig. 3c) or SA (Fig. 3d) over a 72-hour period. To assess the repeatability of induction, engineered cells were alternated between inducer-containing (ON state) and inducer-free (OFF state) media at 24-hour intervals. Consistent

induction and repression profiles were observed over multiple cycles of ON-to-OFF and OFF-to-ON switching with ASA (Fig. 3e) or SA (Fig. 3f). Furthermore, cells retained full responsiveness to vanillic acid (VA), with NLuc expression being dose-dependently inhibited by increasing concentrations of this natural flavoring compound

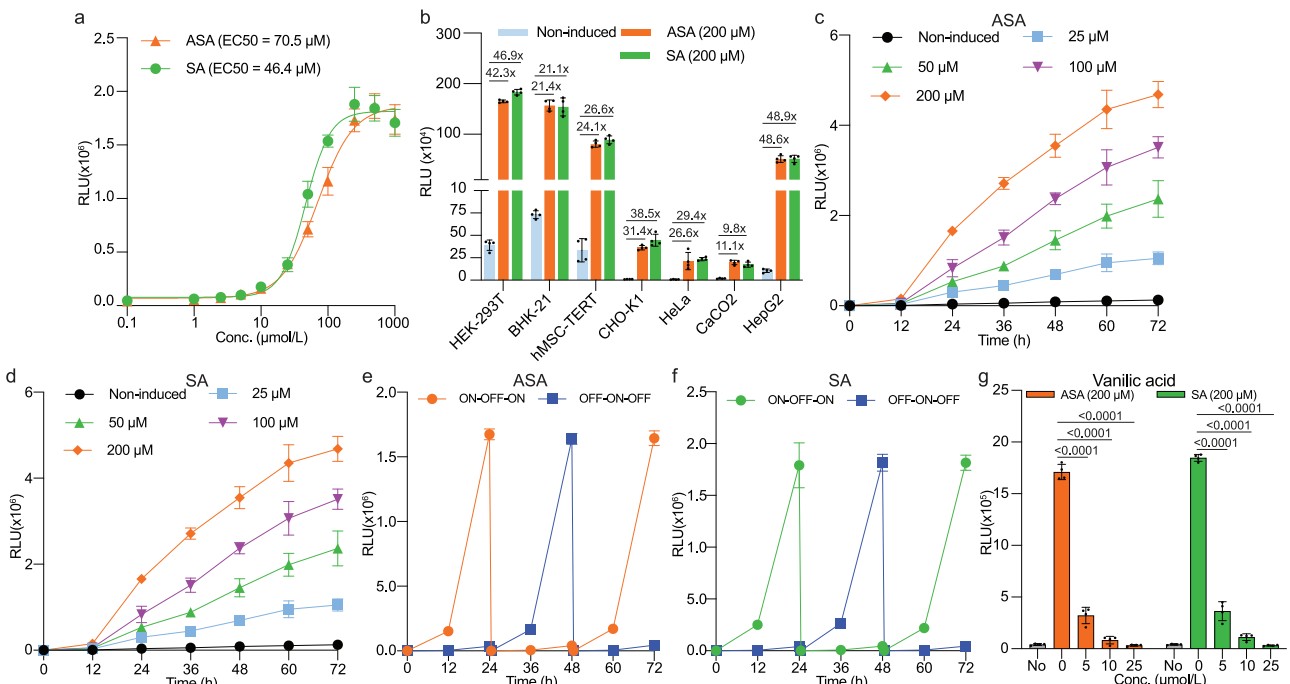

**Fig. 3 | Characterization of the ASA-induced transgene switch.**
**a** Characterization of the dose-response curve of NLuc production by HEK-293T cells transiently cotransfected with the ASPIRIN system. Different concentrations of ASA or SA were applied, and the EC50 value was determined after 24 h of induction. **b** NLuc production in various mammalian cell lines transiently transfected with the ASPIRIN system. Cells were induced with 200 µM ASA or SA, while the control groups received an equivalent volume of vehicle (DMSO). **c, d** Dose-dependent NLuc production kinetics over 72 h in HEK-293T cells transiently transfected with the ASPIRIN components and exposed to ASA (**c**) or SA (**d**) at the indicated concentrations. NLuc levels were quantified in culture supernatants at specified time points. **e, f** Reversibility of HEK-293T cells transiently transfected with the ASPIRIN system. Cells were cultured alternately for 24-hour cycles in medium containing ASA (**e**) or SA (**f**) at 200 µM (ON) or vehicle (DMSO) (OFF). Culture supernatant samples were collected every 12 hours for NLuc production analysis. The culture medium was exchanged and the cell density was re-adjusted every 24 h. **g** Effect of vanillic acid on NLuc reporter expression by ASA- or SA-induced HEK-293T cells transfected with the ASA-controlled gene switch system. Cells were treated with 200 µM ASA or SA, with vanillic acid added at the indicated concentrations. "No" indicates no inducer or vanillic acid added in these groups. The NLuc levels were determined after 24 h. In **b–f**, all induction groups are significantly different from the non-induced groups ($P < 0.05$). Statistical significance was analysed by means of two-sided two-way ANOVA with Tukey's test. All data are means ± SD; $n = 4$. $P$ values were calculated versus the corresponding non-induced control. Source data are provided as a Source Data file.

(Fig. 3g and Supplementary Fig. 9d). This demonstrates the dual-input responsiveness of the ASPIRIN system to two types of small molecules (ASA/SA and VA) and suggests that intervention with VA could provide an additional layer of adjustability and serve as a safety mechanism to transiently override overdosing in emergency situations (Fig. 3g). Importantly, systematic assessment of the influence of ASA and SA on the growth kinetics and recombinant protein production of non-transformed HEK-293T cells (Supplementary Fig. 10), indicated that none of the tested ASA or SA concentrations in the range of 50 to 250 µM had a significant impact on either the rate of cell growth (Supplementary Fig. 10a, c) or the capability for constitutive expression of NLuc (pJH57, $P_{hCMV}$-NLuc-pA) (Supplementary Fig. 10b, d).

### Generation and characterization of an ASPIRIN monoclonal cell line

For a proof-of-principle in vivo study, we focused on T1D, a common chronic condition[45], with insulin as the targeted therapeutic protein. Initially, we established a stable cell line with the ASPIRIN system genomically integrated (Fig. 4a). In this system, the mouse insulin (mINS) coding sequence was positioned downstream of NLuc using a P2A self-cleaving peptide strategy[46] (Fig. 4a and Supplementary Fig. 11a). The system components (pJH230, ITR-$P_{SV40}$-Myr-NPR1-T2$_{mut6}$-pA:$P_{hCMV}$-BlastR-P2A-iRFP-pA-ITR; pJH231, ITR-$P_{SV40}$-NPR4-T2$_{mut6}$-VanR-VP16-pA: $P_{hCMV}$-ZeoR-P2A-mRuby-pA-ITR; pJH2036, ITR-$O_{VanO2}$-$P_{hCMVmin}$-NLuc-P2A-mINS-pA: $P_{RPBSA}$-ECFP-P2A-PuroR-pA-ITR) were semi-randomly integrated into the genome of HEK-293T cells

utilizing the Sleeping Beauty transposase system[47] (Fig. 4a and Supplementary Fig. 11a). Monoclonal cell lines were isolated via fluorescence-activated cell sorting (FACS) (Supplementary Fig. 11b–f) and screened by quantitative NLuc assay after two weeks of culture in selective antibiotic-containing medium (Supplementary Fig. 11g). Subsequently, a secondary screening was conducted to identify the best-in-class cell clone. The most promising cell line, F6, designated as HEK-ASPIRIN (highlighted in blue in Supplementary Fig. 12a), which exhibited the highest induction fold and the lowest basal expression of the NLuc reporter, was selected for further experimentation. FACS analysis confirmed the presence of the three fluorescence-tagged constructs in 99.8% of the analyzed live HEK-ASPIRIN cells (Supplementary Fig. 12b–k). Moreover, to validate the biological activity of insulin expressed by HEK-ASPIRIN, we performed a cell-based insulin activity assay (Supplementary Fig. 13a), in which supernatants from ASA-induced cell cultures were transferred to engineered HEK-293T cells ectopically expressing an insulin receptor and SEAP under an insulin-responsive promoter. The significant increase in SEAP expression unequivocally confirmed the physiological activity of the insulin produced by the established cell line (Supplementary Fig. 13a–e).

Next we compared the transcript levels of *NPR1-T2*, *NPR4-T2*, *VanR-VP16*, *NLuc*, and *mINS* of HEK-ASPIRIN cells incubated with or without inducers (Fig. 4b and Supplementary Data 3). While constitutively expressed genes (*NPR1*, *NPR4*, and *VanR-VP16*) showed similar transcript levels in the induced and non-induced conditions, *NLuc* and *mINS* transcripts were significantly upregulated (> 35 fold) in

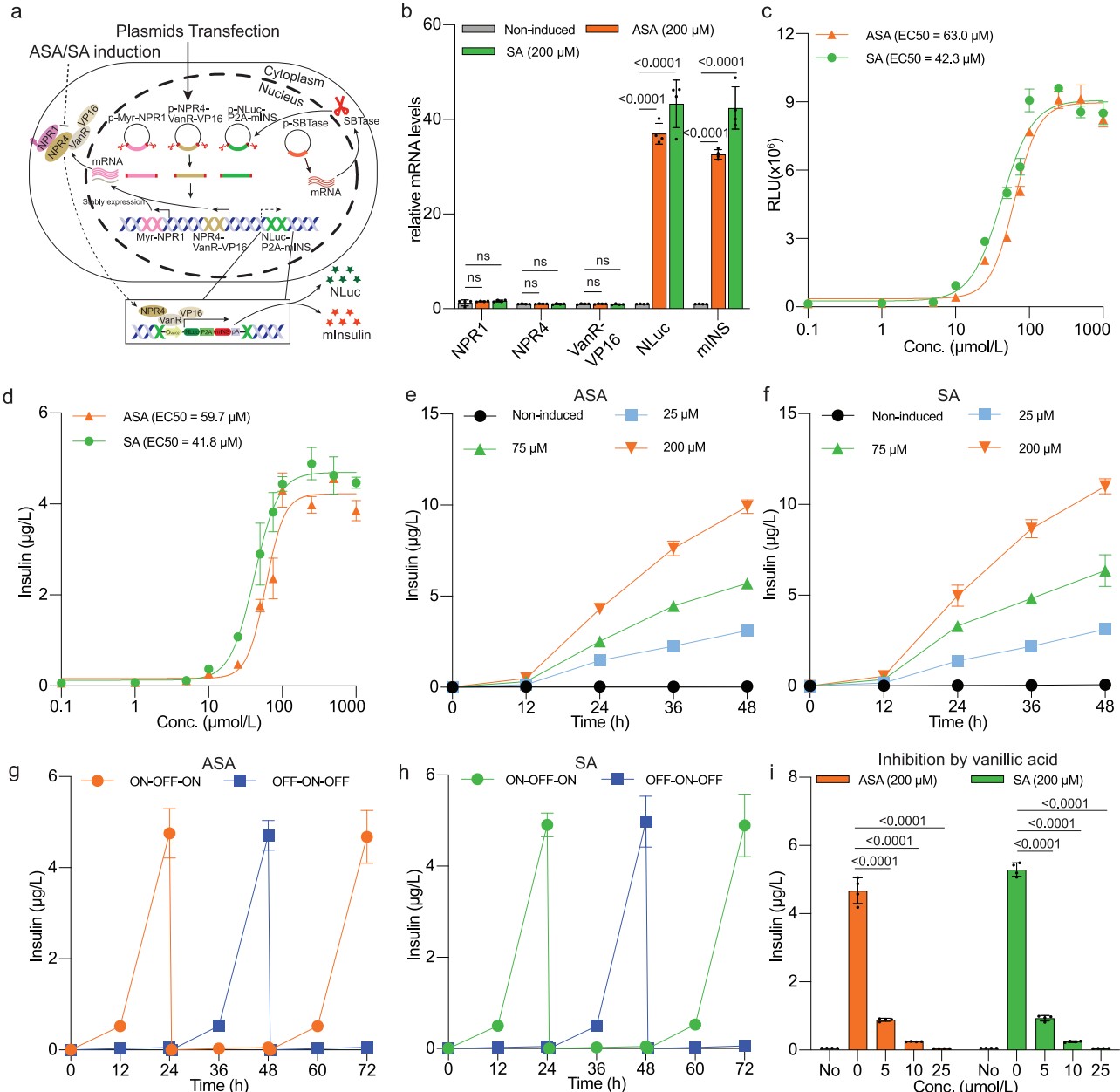

**Fig. 4 | In vitro characterization of the ASPIRIN-engineered monoclonal cell line. a** Schematic representation of stably transfected HEK-293T cells. Cells were co-transfected with plasmids containing the ASPIRIN system flanked by Sleeping Beauty (SB) transposase recognition sites along with a plasmid for expression of the SB transposase. Reporter or therapeutic proteins are labeled as colored stars. **b** Relative mRNA levels of *NPR1, NPR4, VanR-VP16, NLuc* and *mINS* in HEK-ASPIRIN cells in the absence (Non-induced) or presence of ASA or SA (200 μM). **c, d** Characterization of the dose-response curves of NLuc (**c**) and insulin (**d**) in the stable monoclonal cell line HEK-ASPIRIN cultured in the presence of ASA or SA at different concentrations for 24 h. EC50 values are indicated. **e, f** Dose-dependent insulin expression kinetics over 48 h in HEK-ASPIRIN cells exposed to ASA (**e**) or SA (**f**) at the indicated concentrations and time points. **g, h** Reversibility of insulin expression by HEK-ASPIRIN cells

alternately cultured with ASA (**g**) or SA (**h**) at 200 μM (ON) then with the equivalent vehicle (DMSO) (OFF) for 24-h cycles. Culture supernatant samples were collected every 12 hours for analysis of insulin production. The culture medium was exchanged and the cell density was re-adjusted every 24 h. **i**, Effect of vanillic acid on insulin expression by ASA- or SA-induced HEK-ASPIRIN cells. Vanillic acid was added at indicated concentrations to cells treated with 200 μM ASA or SA. "No" indicates no inducer or vanillic acid in these groups. Insulin levels were determined after 24 h. In **e**–**h**, all induction groups differ significantly from the corresponding control groups ($P < 0.05$). Statistical significance was analysed by means of two-sided two-way ANOVA with Tukey's test. All data are means ± SD; $n = 4$. The $P$ values indicate the significance of differences in the mean values versus the non-induced group. Source data are provided as a Source Data file.

cells cultured in ASA or SA-containing medium (Fig. 4b), further confirming the inducibility of the HEK-ASPIRIN cell line. Assessing the cell responsiveness to increasing doses of ASA and SA afforded similar EC50 values for NLuc (Fig. 4c) and insulin (Fig. 4d) expression, consistent with the trend observed in transiently transfected HEK-293T cells (Fig. 3a). Furthermore, HEK-ASPIRIN cells displayed dose- and

time-dependent insulin release profiles over a 48-hour period in response to ASA (Fig. 4e) and SA (Fig. 4f). Reversibility tests showed that HEK-ASPIRIN cells could be activated for insulin expression by ASA or SA addition and subsequently deactivated by washout of ASA (Fig. 4g) or SA (Fig. 4h) at 24-hour intervals, indicating excellent reversibility. Moreover, insulin expression by HEK-ASPIRIN cells in

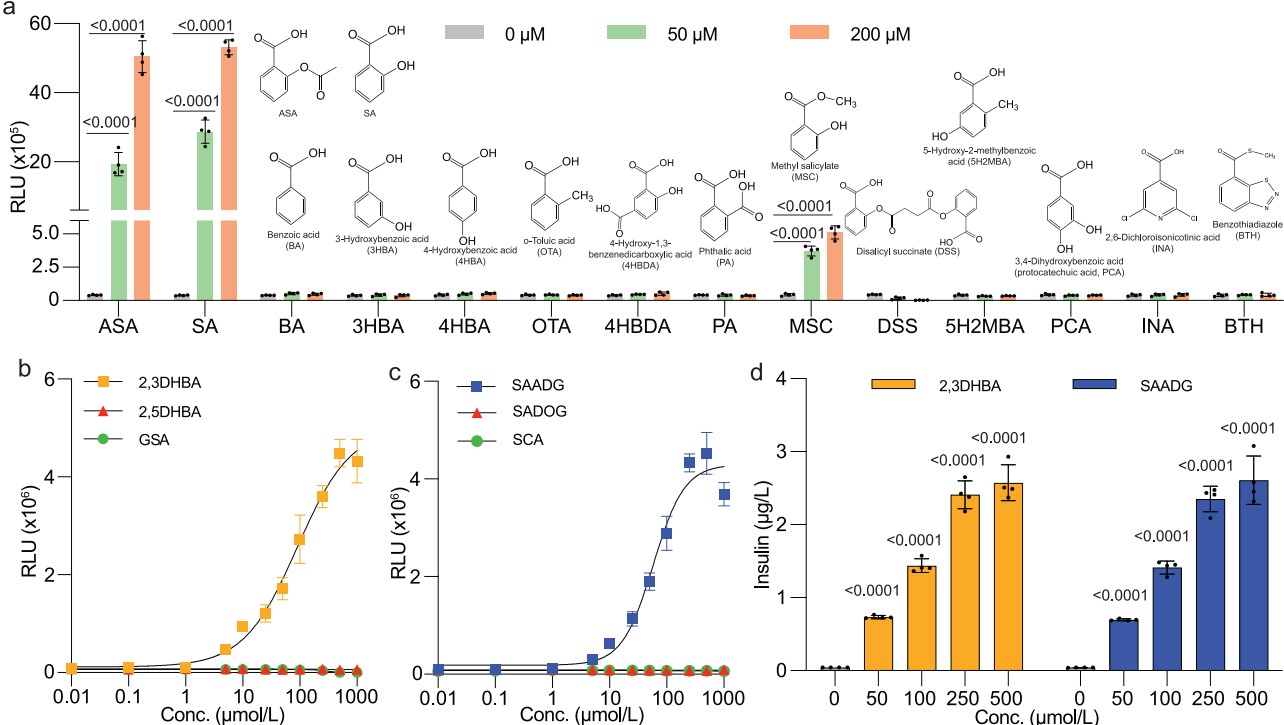

**Fig. 5 | Evaluation of orthogonality and specificity of the ASPIRIN system. a** To evaluate orthogonality and specificity, HEK-ASPIRIN cells were treated with ASA and SA analogs, including benzoic acid (BA), 3-hydroxybenzoic acid (3HBA), 4-hydroxybenzoic acid (4HBA), o-toluic acid (OTA), 4-hydroxy-1,3-benzenedicarboxylic acid (salicylic acid-related compound B, 4HBDA), phthalic acid (PA), methyl salicylate (MSC), disalicyl succinate (DSS), and 5-hydroxy-2-methylbenzoic acid (5H2MBA). **b, c** Dose-response curves were generated for NLuc in HEK-ASPIRIN cells cultured with ASA metabolites 2,3DHBA, 2,5DHBA, gentisuric acid (**b**), and salicylic acid β-D-O-glucuronide (SADOG), salicylic acid acyl-β-D-glucuronide (SAADG) and salicyluric acid (**c**), at various concentrations for 24 h. The ASA metabolic pathway is illustrated in Supplementary Fig. 14b. EC50 values are indicated for 2,3DHBA and SAADG. **d,** The dose-dependent insulin expression profile over 24 h was observed in HEK-ASPIRIN monoclonal cells exposed to 2,3DHBA (brown) and SAADG (pink) at the indicated concentrations. Insulin levels were quantified in culture supernatants after inducer treatment for 24 h. Statistical significance was analysed by means of a two-sided unpaired t test. In **a**, apart from ASA, SA, and MSC, none of the other chemicals showed a significant induction ($P > 0.05$) compared to the non-induced control. All data are means ± SD; $n = 4$. The $P$ values indicate the significance of differences in the mean values versus the non-induced group. Source data are provided as a Source Data file.

response to 200 µM ASA or SA could be dose-dependently shut down by exposure to increasing concentrations of VA (Fig. 4i), consistent with the results in transiently transfected engineered HEK-293T cells (Fig. 3g).

**Specificity and orthogonality of ASPIRIN system**
To assess the specificity of the ASPIRIN system, we screened various ASA analogs as potential inducers (Fig. 5a and Supplementary Fig. 14a). Despite the structural similarities of benzoic acid (BA), 3-hydroxybenzoic acid (3HBA), 4-hydroxybenzoic acid (4HBA), o-toluic acid (OTA), 4-hydroxy-1,3-benzenedicarboxylic acid (4HBDA), phthalic acid (PA), 5-hydroxy-2-methylbenzoic acid (5HMBA), and protocatechuic acid (PCA) to ASA and SA, all these compounds failed to induce NLuc expression in HEK-ASPIRIN cell cultures (Fig. 5a and Supplementary Table 1). Likewise, disalicyl succinate (DSS), formed by the esterification of two molecules of SA with succinic acid, failed to induce the ASPIRIN system (Fig. 5a and Supplementary Fig. 14a). Among the analogs tested, only methyl salicylate (MSC), featuring a methylated carboxyl group in the SA structure, exhibited activity, corresponding to approximately 10% of the fold-induction achieved with ASA or SA (Fig. 5a).

Upon administration to mammalian hosts, ASA undergoes a series of metabolic reactions[33,34]. Initially, esterase digestion converts ASA into SA, which is further metabolized by cytochrome P450 into 2,3-dihydroxybenzoic acid (2,3DHBA) and 2,5-dihydroxybenzoic acid (2,5DHBA/gentisic acid), or conjugated by UDP-glucuronosyltransferases (UGTs) into SA acyl-β-D-glucuronide

(SAADG) and SA β-D-O-glucuronide (SADOG). Additionally, ASA can be converted by acyl-CoA N-acyltransferase into salicyluric acid (SCA), which is further metabolized to gentisuric acid (GSA) (Supplementary Fig. 14b). To assess whether these ASA metabolites can induce the ASPIRIN system, we examined the dose-response relationships in HEK-ASPIRIN cells. Intriguingly, 2,3DHBA effectively induced the system with an EC50 of 90.5 µM (Fig. 5b and Supplementary Fig. 14c), while 2,5DHBA and GSA produced no response (Fig. 5b and Supplementary Fig. 14d, e). Similarly, though SAADG from the UGTs pathway significantly induced the ASPIRIN system with an EC50 of 58.1 µM (Fig. 5c and Supplementary Fig. 14f), SADOG produced no detectable response (Fig. 5c and Supplementary Fig. 14g). The metabolite SCA also produced no detectable response (Fig. 5c and Supplementary Fig. 14h). The responsiveness of the monoclonal cell line to 2,3DHBA and SAADG was also confirmed in terms of insulin production levels (Fig. 5d). These findings suggest that activation of the ASPIRIN system would be sustained for a greater duration in vivo, due to the activity of various ASA metabolites (including SA, 2,3DHBA and SAADG).

**Aspirin-induced insulin expression for the treatment of T1D**
To evaluate HEK-ASPIRIN cells in T1D mice, we encapsulated them in semipermeable alginate beads, which support implanted cell survival by enabling nutrient and oxygen exchange, while protecting from the immune system. We first validated the system in vitro, confirming that

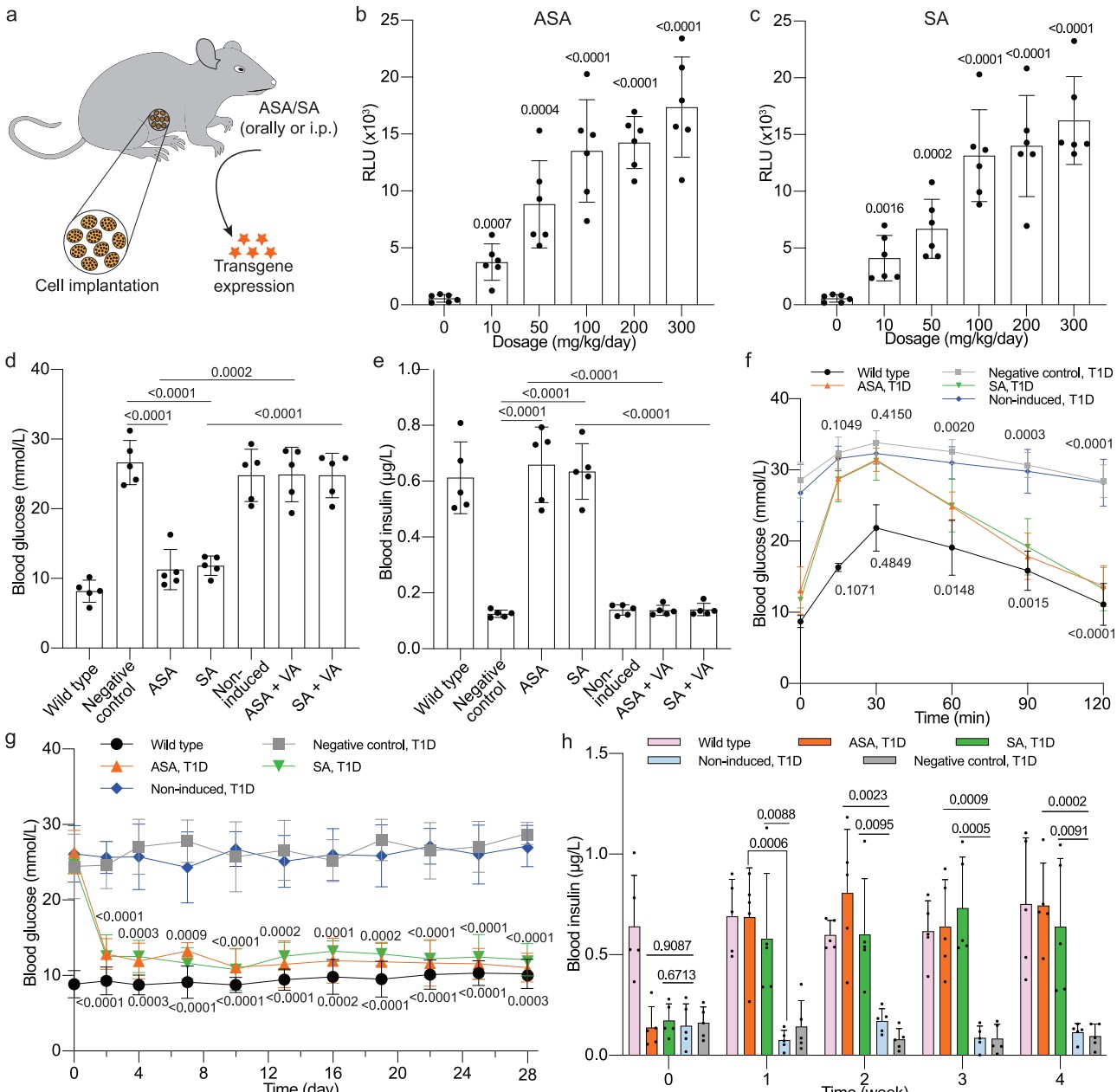

**Fig. 6 | Evaluation and validation of the functionality of the ASPIRIN system in vivo. a** Schematic depiction of how encapsulated engineered HEK-ASPIRIN cells are implanted in mice and induced by ASA or SA. Reporters or therapeutic proteins are labeled as colored stars. **b**, **c** Dose-dependent control of blood NLuc expression induced by ASA (**b**) or SA (**c**). Blood NLuc levels were profiled two days post-implantation of engineered cell implants in WT mice. **d**, **e** Fasting blood glucose (**d**) and insulin (**e**) levels in T1D mice. Insulin levels were quantified after two days of treatment in T1D mice. Mice injected with encapsulated HEK-ASPIRIN cells were either untreated (Non-induced) or treated with ASA or SA (100 mg/kg). Except for the WT, all groups are T1D mice. VA (10 mg/kg) was administered concurrently with the inducer in VA-treated groups. **f** Glucose tolerance test (GTT). GTT was performed by administering 1.5 g/kg aqueous

D-glucose to mice three-day post-implantation of microencapsulated cells following an 8-hour fast. **g** Fasting glycemia was recorded at day 0 and four consecutive weeks after implantation in three groups of T1D mice with HEK-ASPIRIN cell implants induced with ASA, SA, or vehicle (Non-induced). **h** Blood insulin levels were quantified in indicated groups at 0, 1, 2, 3, and 4 weeks post-implantation. In panels **d–h**, WT and T1D mice without any implant or treatment were utilized as controls. All data are presented as means ± SD; in panels **b**, **c**, *n* = 6; in panels **d–h**, *n* = 5. Statistical significance was analysed by means of a two-sided unpaired t test (**b**, **c**), one-way (**d**, **e**) or two-way ANOVA (**f–h**). In **f**, **g**, the *P*-values denote the significance of differences in mean values of ASA (first row) and SA (second row)-treated groups versus the non-induced control groups. Source data are provided as a Source Data file.

encapsulated cells dose-dependently secreted NLuc and insulin into the culture medium in response to the inducers (Supplementary Fig. 15). For initial in vivo assessment, wild-type (WT) mice were implanted intraperitoneally (i.p.) with encapsulated cells and then treated with varying concentrations of ASA or SA (Fig. 6a). Blood analysis revealed a dose-dependent increase of NLuc in response to ASA (Fig. 6b) or SA (Fig. 6c) administration, confirming that transgene

expression in implanted cells can be precisely regulated by these compounds.

To investigate therapeutic protein regulation, we implanted encapsulated HEK-ASPIRIN cells i.p. in T1D mice. Inducer administration significantly reduced fasting blood glucose to WT levels (Fig. 6d), while markedly increasing circulating insulin (Fig. 6e). Furthermore, co-administration of VA to the ASA- or SA-treated groups reversed

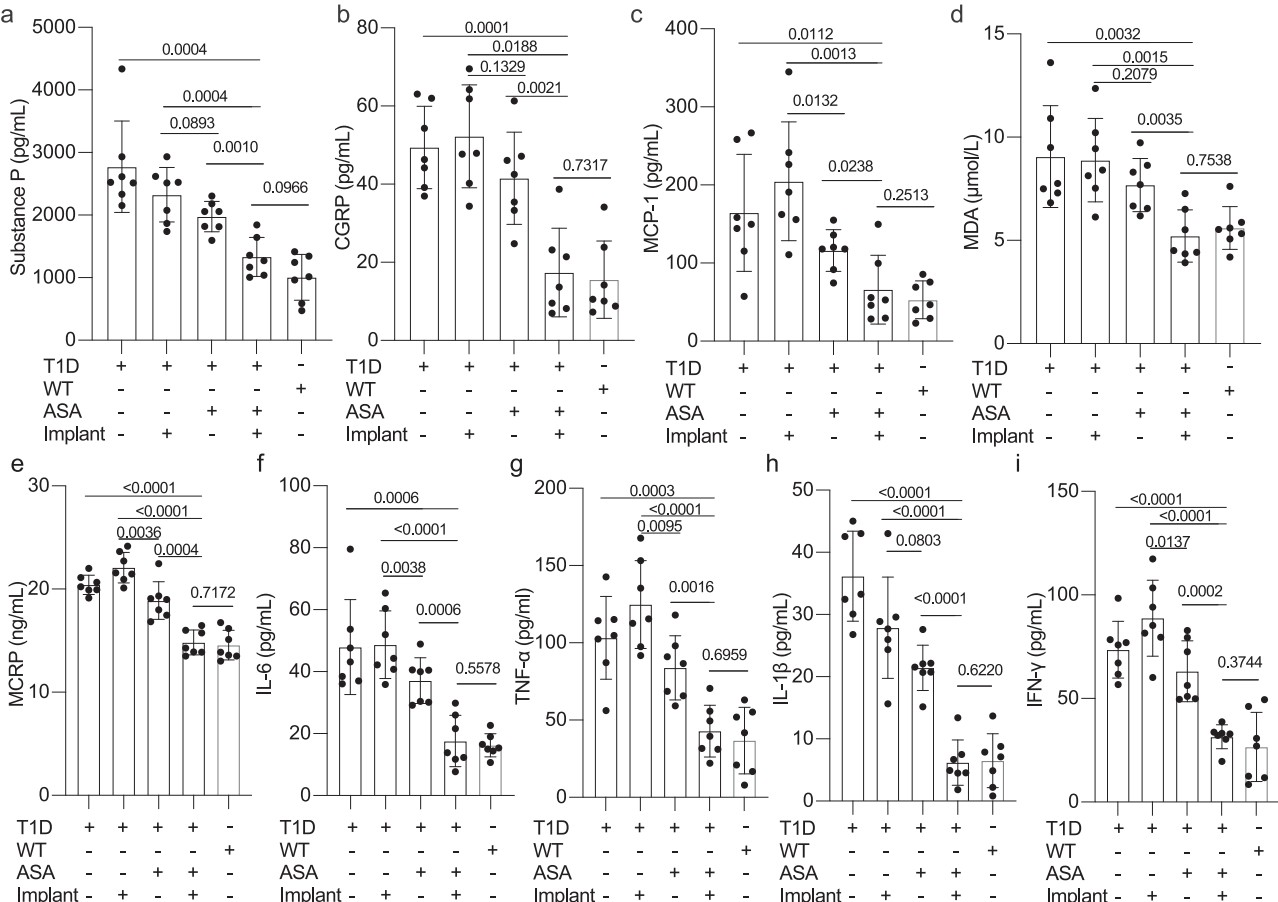

**Fig. 7 | Evaluation of anti-inflammatory activity of ASA in vivo. a, b** Analysis of biomarkers pertaining to pain perception in T1D mice: neuropeptide substance P (**a**) and calcitonin gene-related peptide (CGRP) (**b**). **c–i** Profiling of inflammation biomarkers of diabetic complications in serum of mice. Levels of MCP-1 (**c**), MDA (**d**), MCRP (**e**), IL-6 (**f**), TNF-α (**g**), IL-1β (**h**) and IFN-γ (**i**) were measured in the serum of WT without cell implants, T1D negative control with or without cell implants, stimulated (ASA (+))T1D mice with or without cell implants. All T1D mice used in this study were i.p. administered with ASA (100 mg/kg/day) or vehicle twelve weeks post streptozotocin (STZ) injection. Statistical significance was analysed by means of a two-sided unpaired t test. All data are presented as means ± SD; $n = 7$. The P values denote the significance of differences in mean values versus the non-induced control or indicated groups. Source data are provided as a Source Data file.

these effects, restoring both fasting blood glucose (Fig. 6d) and insulin (Fig. 6e) to non-induced levels. By day 3 post-implantation, an intra-peritoneal glucose tolerance test (ipGTT) confirmed that ASA- or SA-treated mice effectively restored blood glucose homeostasis and mitigated postprandial fluctuations compared to control groups lacking implants or carrying non-induced implants (Fig. 6f). Additionally, mice receiving ASA and SA treatment exhibited lower fasting glycemia levels, maintaining normoglycemia throughout the four-week study, comparable to WT levels (Fig. 6g). Weekly insulin profiling further confirmed a significant increase in blood insulin in ASA- and SA-induced mice relative to the non-induced controls (Fig. 6h).

### ASA alleviates diabetic neuropathy and chronic inflammation-related biomarkers

Given the natural analgesic function of ASA, it may be particularly suited for regulation of therapeutic transgene expression as a part of anti-nociceptive combination therapies. First, to confirm the anti-nociceptive efficacy of ASA, we conducted three standard nociceptive behavioral tests in WT mice. Firstly, in the hindpaw formalin injection test[48,49], WT mice were injected with a diluted formaldehyde solution into the right hindpaw, and the time spent licking or biting the injected paw was measured in mice treated with or without oral administration of ASA. Our findings revealed that ASA significantly reduced the licking time in a dose-dependent manner, both during the "early phase"

(0–10 min) and the "late phase" (10–40 min) after formalin injection (Supplementary Fig. 16a). We also employed the acetic acid-evoked writing test[48,49], in which an intraperitoneal injection of acetic acid induces a characteristic writhing response in mice, indicative of visc-eral pain. The frequency of these writhes serves as a measure of the effectiveness of analgesic treatments. In our experiment, after ASA administration, mice were intraperitoneally injected with acetic acid, and we quantified the number of abdominal constrictions over a 30-min period. Pretreatment of the mice with ASA at doses ranging from 25 to 300 mg/kg exhibited anti-nociceptive activity, as evidenced by a dose-dependent inhibition of abdominal constrictions (Supple-mentary Fig. 16b). Furthermore, we performed the thermal-induced tail-flick test[49,50], in which the mouse's tail is immersed in a warm water bath to measure the tail withdrawal latency. Oral ASA administration resulted in a dose-related increase in the latency of tail withdrawal, compared to the control group injected with vehicle only (Supple-mentary Fig. 16c).

Next, to create an experimental model of diabetic neuropathy, we maintained high blood glucose levels in STZ-induced T1D diabetic mice (> 20 mmol/L) for a minimum of twelve weeks[51–53] (Supplementary Fig. 17a) and found that there was indeed a positive correlation between hyperglycemia and systemic levels of inflammatory bio-markers of nociception, such as neuropeptide substance P (SP) and calcitonin gene-related peptide (CGRP)[54,55] (Fig. 7a, b). To assess the

therapeutic efficacy of the ASPIRIN system in treating such late-stage diabetic complications in vivo, we implanted alginate-encapsulated HEK-ASPIRIN cells intraperitoneally (i.p.) in these mice. A five-day treatment regimen with ASA not only significantly ameliorated fasting glycemia (Supplementary Fig. 17b), but also normalized the expression of pain perception-associated biomarkers in the implanted T1D mice (Fig. 7a, b). Similarly, serum levels of monocyte chemoattractant protein-1 (MCP-1) and malondialdehyde (MDA)[55,56], two biomarkers of diabetic complications, were also restored in a ASA- and ASPIRIN-dependent manner (Fig. 7c, d).

Lastly, we quantified systemic biomarkers of chronic inflammation in both WT and late-stage T1D mice[55,57,58], including C-reactive protein (CRP) (Fig. 7e), interleukin-6 (IL-6) (Fig. 7f), tumor necrosis factor-alpha (TNF-alpha) (Fig. 7g), interleukin-1 beta (IL-1β) (Fig. 7h) and interferon-gamma (IFN-γ) (Fig. 7i). These inflammatory biomarkers were all significantly upregulated in untreated T1D mice or T1D receiving HEK-ASPIRIN but no ASA, whereas T1D mice implanted with HEK-ASPIRIN and treated with ASA exhibited a remarkable reduction to levels akin to those observed in healthy controls (WT, ASA(-)) (Fig. 7e–i). These results demonstrate that aspirin alone has minimal effects on biomarkers associated with diabetic complications. However, with the assistance of the ASPIRIN system, aspirin significantly downregulates these biomarkers to wild-type levels (Fig. 7a–i). Collectively, these findings support the idea that ASPIRIN could become an attractive strategy for therapeutic transgene regulation by elegantly exploiting the treatment benefits of ASA itself.

## Discussion

We have developed a sensitive, reversible, and tunable transgene platform that enables the expression of therapeutic proteins in mammalian cells in response to ASA (aspirin), a widely used medication for treating fever, inflammation, pain, and cardiovascular diseases. Leveraging the plant-derived SA sensors NPR1 and NPR4[35,36], we screened a set of truncation variants of NPR1 and NPR4 and further evolved them via error-prone PCR-based random mutagenesis/screening to provide a sensitive ASA sensor that forms the basis of the ASPIRIN system. Notably, most chemical compounds structurally similar to ASA failed to activate the ASPIRIN system, which was only responsive to a narrow spectrum of metabolites derived from the metabolism of ASA in mammals, including SA, 2,3DHBA and SAADG[33,34]. This feature prolongs therapeutic protein expression in vivo in response to ASA and consequently prolongs the system's therapeutic effect. Importantly, the ASPIRIN system is a platform that should also be applicable to express other therapeutic proteins, serving to combine the benefits of a well-established small-molecule drug with gene- or cell-based therapies.

Key considerations in developing a safe gene switch/inducer system include specificity, reversibility, cytotoxicity, pharmacokinetics, biocompatibility, and long-term effects[1,7,8,11,21–23]. The ASPIRIN system has undergone extensive engineering to respond precisely and specifically to ASA and its metabolites, and provides robust specificity and reversibility across multiple cycles of activation and deactivation. Moreover, given the longstanding global use of ASA in treating fever, inflammation, pain, and cardiovascular diseases for over a century[27,30], the remaining concerns mentioned above are inherently addressed. Emerging evidence suggests that regular ASA usage may also mitigate the risk of certain cancers, notably colorectal cancer[30,31,59], through its anti-inflammatory properties and inhibition of abnormal cell growth, as well as having potential for managing neurodegenerative disorders and pregnancy-related issues such as preeclampsia[29].

Given these multifaceted therapeutic benefits of ASA itself, considerable efforts have been made to engineer a mammalian ASA-induced gene switch, with initial forays dating back to bacterial studies in 2007[42]. Subsequent endeavors have explored ASA-induced systems in depth[39–42,60], but despite these efforts, the realization of a practical

system in mammalian cells has remained elusive. A recent attempt by the Cirino group focused on incorporating *MarR* into Jurkat cells[60]. However, induction of the EGFP reporter, albeit significant, was observed only at high salicylic acid concentration (2 mM), tenfold greater than that employed in our study, and this afforded only a marginal 1.3x induction fold.

Protein structure and domain organization analysis are crucial in protein engineering[43,44]. Indeed, our findings revealed that full-length NPR4 paired with either full-length NPR1 or truncated versions exhibited only minimal induction in response to ASA or SA. However, the complex of NPR1 and NPR4 with truncation 2 (NPR4-T2) together with an elongated salicylic acid-binding core (SBC) showed a notable improvement in induction factor. Based on these results, we selected the best-performing combination, NPR1-T2 and NPR4-T2, for further optimization. Directed evolution using error-prone PCR, which mimics natural selection[61,62], was employed to generate a random mutant library. After multiple rounds of high-throughput iterations and screening, we successfully reduced baseline expression while simultaneously enhancing the expression level, resulting in a significant improvement in induction fold. To minimize the potential effects of basal expression, we selected the best-in-class *NPR1-T2* and *NPR4-T2* mutant variants for construction of our gene switch. Importantly, we found that all the mutations of the top-performing *NPR4* are clustered within the SA binding pocket[35], suggesting that artificial natural selection was indeed achieved to enhance the sensitivity of NPR4 to ASA. Additionally, the enhanced sensitivity of *NPR1-mut* and *NPR4-mut* to SA might support potential applications to increase disease resistance of crops and to improve yields in plant synthetic biology[36,63].

For a proof-of-concept validation, we focused on T1D and its associated complications in a mouse model of late-stage T1D. Diabetes mellitus can lead to a variety of secondary complications, especially when blood sugar levels are poorly controlled over an extended period. Chronic inflammation plays a crucial role in the development and progression of many diabetic complications, including cardiovascular diseases, peripheral neuropathy, diabetic nephropathy, diabetic retinopathy, skin and foot problems and so on[52–56,58,64]. Our results demonstrate that the developed ASPIRIN system not only effectively regulated expression of the therapeutic protein, insulin, in diabetic mice, restoring normoglycemia, but also reduced the levels of biomarkers related to pain and diabetic inflammation via the natural action of ASA. Thus, the inherent advantages and accessibility of ASA substantialy enhance the efficacy of our ASA-induced transgene switch, laying the foundation for the development of a range of cell-based combination gene therapies based on the ASPIRIN system, which could potentially be used to treat a range of conditions, including heart attacks, strokes, coloral cancer and neurodegenerative diseases in the future.

## Methods
### Ethical statement
This study was conducted in strict accordance with all relevant ethical regulations and animal welfare legislation in France or China. The experiments were approved by the French Republic (project no. DR2018-40v5 and APAFIS no. 16753), and conducted by Jinbo Huang, Shuai Xue, and Ghislaine Charpin-El Hamri (no. 69266309) at the University of Lyon, Institut Universitaire de Technologie (IUT, F69622 Villeurbanne, France), or approved by the Institutional Animal Care and Use Committee (IACUC) of Westlake University and in accordance with the Animal Care Guidelines of the Ministry of Science and Technology of the People's Republic of China, and performed by Ting Gao and Shuai Xue according to the protocol (Protocol ID: AP#24-088-XMQ). To minimize variability associated with hormonal fluctuations in females, we used only male mice. This approach aligns with previous studies in the field[17,65–67] and allows for a more controlled investigation of diabetes. Mice were housed under a 12-hour light–dark cycle, with

five mice per cage. The ambient temperature was maintained at $21 \pm 1\,°C$, with a humidity level of $50 \pm 10\%$.

## Key plasmids used in this study

All expression vectors were using Benchling (www.benchling.com), with construction details available in Supplementary Data 1-2. Target DNA fragments were amplified via PCR using Q5® High-Fidelity 2x Master Mix (M0492L, New England BioLabs), and purified by gel electrophoresis, followed by DNA extraction with a Zymo Research recovery kit (D4002). Backbone plasmids were digested with New England BioLabs restriction enzymes, and inserts were assembled using the Gibson assembly protocol (E2611L, New England BioLabs). For transformation, 50 µl of competent XL10 gold K12 *Escherichia coli* (Stratagene) was mixed with the ligation reaction, incubated on ice for 15 min, and heat-shocked at $42\,°C$ for 90 seconds. Transformed bacteria were plated on Lurie-Bertani (LB)-agar plates containing the appropriate antibiotics for selection. Individual colonies were cultured in LB broth supplemented with antibiotics, and plasmid DNA was extracted using either the Zyppy Plasmid Miniprep Kit (D4037, Zymo Research) or the ZymoPURE II Plasmid Midiprep Kit (D4201, Zymo Research). The constructs were sequenced verified by Microsynth AG (Balgach, Switzerland).

## Domain analysis

The truncations of NPR1 and NPR4 proteins were designed based on analysis of the sequences from the NCBI database (https://www.ncbi.nlm.nih.gov/Structure/cdd/wrpsb.cgi) with the aid of a structure prediction platform (https://www.compbio.dundee.ac.uk/jpred/).

## Cell culture and transfection

The following cell lines were used in this study: human embryonic kidney cells (HEK-293T, ATCC: CRL-11268), baby hamster kidney cells (BHK-21, ATCC: CCL-10), Chinese hamster ovary cells (CHO-K1, ATCC: CCL-61), human telomerase-immortalized mesenchymal stem cells (hMSC-TERT)[68], human cervical adenocarcinoma cells (HeLa, ATCC: CCL-2), human colorectal adenocarcinoma cells (Caco-2, ATCC: HTB-37), and human liver cancer cells (Hep G2, ATCC: CRL-11997). All cell lines were maintained in Dulbecco's Modified Eagle's medium (DMEM, cat. no. 52100-39, Thermo Fisher Scientific) supplemented with 10% fetal bovine serum (FBS, cat. no. F7524, lot no. 022M3395, Sigma-Aldrich) and 1% (v/v) penicillin-streptomycin solution (cat. no. L0022, Biowest). The cultures were incubated at $37\,°C$ in a humidified environment with 5% $CO_2$. For transfection, 15,000 cells per well were seeded in 96-well plates (cat. no. 3599, Corning Inc. Life Sciences) and incubated for 24 h to allow attachment. A transfection mixture was prepared with 0.5 µg polyethyleneimine (PEI MAX®, MW 40,000, stock solution 1 µg/µL in ddH$_2$O, cat. no. 24765-2, Polysciences) and 0.125 µg of plasmid DNA, maintaining equimolar ratios for plasmid combinations. A total of 40 µL of this mixture was added per well. After 12 h, the medium was replaced with 100 µL of fresh culture medium. All cell lines were authenticated by the supplier and the authorities of the ETH Zurich in Basel, Switzerland.

## Random mutagenesis of NPR1 and NPR4

To improve the responsiveness of NPR1 and NPR4 constructs to ASA, iterative rounds of random mutagenesis were carried out using the GeneMorph II random mutagenesis kit (200552, Agilent Technologies) following the manufacturer's protocol. Briefly, mutant libraries of NPR1 and NPR4 were generated via PCR amplification, purified and inserted into pJH140 (digested with *Spe*I and *Bam*HI) or pJH196 (digested with *Eco*RI and *Spe*I), respectively. The resulting libraries were transformed and individual clones were isolated and grown in 96-deep-well plates at $37\,°C$ overnight. For plasmid extraction, we used the Zyppy-96 Plasmid Kit (D4042, Zymo Research). To evaluate the functionality of the generated mutants, selected clones were

transfected into HEK-293T cells plated in 96-well plates (cat. no. 3599, Corning Inc. Life Sciences). Twenty-four hours post-induction with ASA, activation efficiency was assessed. Multiple rounds of mutagenisis were performed to further refine the system.

## Monoclonal cell line generation and fluorescence-based sorting

To develop the ASPIRIN$_{INS}$ stable cell line, HEK-293T cells ($3.5 \times 10^5$) were plated in 6-well plates and allowed to adhere for 24 h. Cells were then co-transfected with 1200 ng of pJH230, 300 ng of pJH231, 1200 ng of pJH2036, and 300 ng of pJH42 (P$_{hCMV}$-SB100X-pA), the latter encoding the Sleeping Beauty transposase SB100X for genomic integration. After 24 h, we exchanged the medium to a fresh medium supplemented with puromycin (1 µg/ml), blasticidin (10 µg/ml) and zeocin (50 µg/ml) for a three-day antibiotic selection period. The surviving polyclonal cell population was subjected to fluorescence-activated cell sorting (FACS) using a BD Biosciences system. Sorting was based on fluorescent marker expression: yPET (517/530), mRuby (558/605), and iRFP (690/713). Cells exhibiting triple-positive fluorescence were isolated as single clones in 96-well plates and maintained in antibiotic-containing medium. After two weeks of expansion, individual clones were tested for ASA responsiveness, and the highest-performing monoclonal cell was selected for downstream studies. Flow cytometry analysis confirmed stable triple-positive fluorescence expression.

## Impact of ASA and SA on cell proliferation

To evaluate cell growth kinectics, $4.0 \times 10^4$ cells per well were plated in 24-well plates and allow to adhere for 24 h. Cells were then treated with defined concentrations of ASA and SA, while control groups received an equivalent volume of DMSO as a vehicle control. Cell viability was monitored at multiple time points over a 48-h period to assess proliferation changes.

## Influence of ASA and SA on recombinant protein productivity

To assess protein expression levels, HEK-293T cells ($2.0 \times 10^6$) were plated in 10 cm dishes and incubated for 24 h to allow adhesion. Cells were then transfected with 30 µg of pJH57 (P$_{hCMV}$-NLuc-pA) for 12 h, resuspended, and evenly distributed into two 96-well plates. Experimental groups were treated with specific concentrations of ASA and SA, while control groups received an equivalent volume of DMSO. Samples were collected at designated time points over 48-h period to measure NLuc production.

## Chemicals used in this study

ASA and related compounds used in this study were from Sigma-Aldrich unless otherwise stated. Details are given in Supplementary table 1. ASA and its analogs used in this study were solubilized in 50% DMSO for in vitro assessments. For animal experiments, ASA and SA were freshly prepared in saline solution containing 0.5% (w/v) carboxymethylcellulose (cat. no. 21902-100 G, Sigma-Aldrich).

## Chemical structure drawing

Chemical structures were drawn using ChemDraw (v20.0.0.38, PerkinElmer).

## Cell viability

To assess viability, cells were incubated with resazurin (50 µg/mL, cat. no. R7017, Sigma-Aldrich) for 2 h at $37\,°C$, and the fluorescence was measured at 540/590 nm (Tecan SPARK plate reader, Tecan Group AG). Relative cell viability was calculated by normalizing the fluorescence intensity of the non-stimulated control group to 100%.

## SEAP activity measurement

SEAP concentration in cell culture supernatants was determined using a colorimetric assay. Briefly, 80 µL of supernatant was heat-inactivated

at 65 °C for 30 minutes, then mixed with 100 μL of 2× SEAP assay buffer (20 mM homoarginine, 1 mM $MgCl_2$, and 21% diethanolamine, pH 9.8). Subsequently, 20 μL of a substrate solution containing 120 mM p-nitrophenyl phosphate (cat. no. AC128860100, Thermo Fisher Scientific) was added. Absorbance at 405 nm was recorded over 30 min at 37 °C using a Tecan SPARK plate reader.

## Nanoluciferase quantification
NLuc levels in cell culture supernatants were measured using the Nano-Glo Luciferase Assay System (cat. no. N1110, Promega, Wisconsin, USA) as per the manufacturer's guidelines. A 7.5 μl aliquot of culture supernatant was mixed with 7.5 μl of buffer-substrate solution (prepared at a 50:1 ratio) in a 384-well plate (cat. no. 781076, Greiner BioOne, Kremsmünster, Austria). The plate was centrifuged at 3000 g for 1 min before luminescence readings were recorded using a Tecan Spark plate reader (Tecan Group Ltd., Switzerland).

## Co-IP and WB analysis
To analyze protein interactions and expression levels, HEK-293T cells were co-transfected with constructs encoding 3×HA tagged NPR1-T2 and and 3×FLAG-tagged NPR4-T2-VanR-VP16. After 24 h, cells were lysed, and protein extracts were prepared. Co-IP was carried out using the Pierce™ HA-Tag Magnetic IP/Co-IP Kit (Thermo Fisher, cat. no. 88838) following the manufacturer's protocols. Total protein content was quantified using the Pierce BCA Protein Assay Kit (Thermo Fisher, cat. no. 23225). For WB analysis, samples were denatured in 2× Laemmli buffer at 95 °C for 5 min, and 10 μl of each sample was subjected to SDS–PAGE. Proteins were transferred onto a polyvinylidene fluoride (PVDF) membrane (Bio-Rad, cat. no. 1704156) and blocked with 5% nonfat milk in TBST for 1 h at room temperature. For FLAG detection, membranes were incubated overnight at 4 °C with a mouse anti-FLAG primary antibody (Sigma-Aldrich, cat. no. F1804, 1:2000), followed by an anti-mouse HRP-conjugated secondary antibody (Abcam, cat. no. ab205719, 1:2000). Signals were visualized using the Pierce ECL Western Blotting Substrate (ThermoFisher, cat. no. 32106) and a chemiluminescence detection system (Azure 400, Azure Biosystems, USA). For HA detection, membranes were stripped, re-blocked and incubated overnight at 4 °C with a rabbit anti-HA primary antibody (Abcam, cat. no. ab236632, 1:1000), followed by an anti-rabbit HRP-conjugated secondary antibody (Sigma-Aldrich, cat. no. A6154, 1:5000). Chemiluminescence detection was performed as described above.

## Fluorescence microscopy
Cells were seeded in 24-well plates with clear, flat bottoms (Greiner bio one, lot # E18053JY) and allowed to adhere overnight before transfection. After 12 h, cells were treated with inducers or vehicle controls and incubated for an additional 24 h before imaging. Hoechst 33342 (cat. no.: H3570, ThermoFisher) was added at a final concentration of 1 μg/ml to stain nuclei. Imaging was carried out using a Leica SP8 confocal microscope equipped with a laser excitation system. Hoechst 33342 fluorescence was excited at 380 nm, with emission collected between 410 and 430 nm. EGFP and mCherry fluorescence were analyzed at their respective excitation and emission wavelengths (488 nm/ 510 nm for EGFP and 587 nm/610 nm for mCherry). Fluorescence intensity was quantified using ImageJ software (1.54 g, Java13.0.6, NIH, USA).

## Insulin activity bioassay
To assess the biological activity of recombinant insulin, a cell-based assay was performed. Engineered HEK-293T cells stably expressing the insulin receptor ($P_{hCMV}$-hIR-pA) alongside pMF111 ($P_{TRE}$-SEAP-pA) and Mkp37 ($P_{hCMV}$-TetR-Elk1-pA) were incubated for 36 h with conditioned medium containing secreted insulin. Activation of the insulin receptor initiates a MAPK signaling cascade, leading to SEAP expression. SEAP activity in the culture supernatant was quantified as described above.

## RNA sample extraction and quantitative PCR (qPCR) assay
HEK-293T cells were seeded in 10-cm dishes at a density of $3.0 \times 10^6$ cells and cultured overnight. After 24 h of treatment with 200 μM ASA, SA, or DMSO (vehicle), cells were harvested for RNA extraction using the Quick-RNA Miniprep Kit (Zymo Research, cat. no. R1054). RNA concentration was measured using a NanoDrop 2000 (Thermo Fisher) and adjusted to 100 ng/μl. cDNA synthesis was carried out using the High-Capacity cDNA Reverse Transcription Kit (Applied Biosystems, cat. no. 4368814) according to the manufacturer's protocol. The resulting cDNA (diluted 1:5) was mixed with SYBR Green Supermix (Bio-Rad, cat. no. 1725271) for subsequent qPCR analysis on the QuantStudio 3 system (Thermo Fisher). The *glyceraldehyde 3-phosphate dehydrogenase* (*GAPDH*) served as the house-keeping control. Primer sequences used for qPCR are detailed in Supplementary Data 3.

## Microencapsulation and implantation of monoclonal HEK-ASPIRIN cells
To protect engineered HEK-ASPIRIN cells from immune rejection while allowing nutrient and protein diffusion, we used a clinically approved alginate-based encapsulation method. Briefly, $5.0 \times 10^7$ cells were harvested, washed with MOPS buffer (10 mM MOPS 7.2, 150 mM NaCl), and mixed with an alginate solution (1.6% w/v, Nadium alginate, cat. no. 11061528, Buechi Labortechnik AG, Switzerland) and a poly-L-lysine (PLL) solution (0.05% w/v; PLL 2000: cat. no. 25988-63-0, Alamanda Polymers Inc., USA). Encapsulation was performed using an Inotech Encapsulator IE-50R (EncapBiosystems Inc.), forming microcapsules (approximately 400 μm diameter) under the following conditions: a 200 μm nozzle, a vibration frequency of 1000 Hz, a 20 mL syringe with a flow rate of 20 mL/min, and a bead dispersion voltage of 1200 V for bead dispersion. Encapsulated cells were cultured in DMEM medium for 24 h before implantation. The medium was replaced with serum-free DMEM, and 1.0 mL of the encapsulated beads containing $5 \times 10^6$ cells, were intraperitoneally injected into mice using a standard 3 mL syringe (cat. no. 9400038, Becton Dickinson) with a 0.7 × 30 mm needle (cat. no. 30382903009009, Becton Dickinson).

## Blood glucose and insulin analsyis
Blood was drawn from tail or saphenous veins using a 20 μL glass micro-hematocrit capillary (Avantor® VWR, cat. no. 521-9100) and transferred into BD Microtainer® blood collection tubes (cat. no.: BDAM365968). The samples were then centrifuged at 8000 × g for 2 min, and the serum was either immediately analyzed or stored at -80 °C within 1 h of collection. Glucose levels were measured using clinically licensed Contour®Next test strips and the Contour®Next ONE reader (Ascensia Diabetes Care, Switzerland). Insulin was quantified by an ELISA kit (cat. no. 10-1247-01, Mercordia).

## Assay of serum inflammatory biomarkers
The concentrations of inflammatory biomarkers, including IFN-γ, IL-6, IL-1β, TNF-α, MCRP, MDA, MCP-1, CGRP and SP, were quantified in mouse serum using specific enzyme-linked immunosorbent assay (ELISA) kits. The following kits were utilized: IFN- γ mouse ELISA kit (cat. no. ab282874, Abcam), IL-6 mouse ELISA kit (cat. no. ab100712, Abcam), IL-1β mouse ELISA Kit (cat. No. BMS6002, ThermoFisher), TNF-α mouse ELISA kit (cat. no. BMS607HS, ThermoFisher), MCRP ELISA Kit (cat. no. ab157712, Abcam), malondialdehyde (MDA) colorimetric assay kit (cat. no. EEA015, ThermoFisher), mouse MCP1 ELISA Kit (cat. no. ab100721, Abcam), mouse CGRP ELISA Kit (cat. no. A76318, ANTIBODIES) and mouse substance P ELISA Kit (cat. no. A80235, ANTIBODIES).

## Preparation and confirmation of T1D mouse model
T1D was induced in eight-week-old male WT C57BL/6 J mice (Janvier Labs) via intraperitoneal administration of streptozotocin (STZ,

60 mg/kg; cat. no. S0130, Sigma-Aldrich) in sodium citrate buffer (pH 4.3, 100 μL). Mice were injected daily for five consecutive days following an 8-h fasting period. One week after the final injection, fasting blood glucose levels were measured to confirm hyperglycemia. Glucose tolerance was assessed by administering 1.5 g/kg glucose intraperitoneally and monitoring blood glucose levels at defined intervals.

### Anti-nociceptive activity of ASA in diabetic mice

For the assessment of the anti-nociceptive activity of ASA in diabetic mice, 10-week-old male WT and STZ-induced T1D mice were utilized. (1) Measurement of formalin-induced hindpaw licking behavior. Mice were randomly divided into groups and orally administered ASA at specified concentrations for 15 min. Subsequently, formalin (10 μl/mice, 5% v/v in saline solution) was injected intraplantarly into the right hindpaw. Immediately after injection, the mice were placed in transparent plexiglass chambers and their behavior was recorded using a digital camera. The duration of hindpaw licking was monitored and counted during the "early phase" (0–10 min) and "late phase" (10–30 min) after formalin injection. (2) Assessment of acetic acid-evoked abdominal constriction. The acetic acid-induced abdominal constriction assay was conducted as previously described[48]. Briefly, mice were orally administered ASA at specified dosages for 15 min. Subsequently, freshly prepared acetic acid (200 μl/mice, 2.5% v/v in saline) was injected intraperitoneally. The mice were immediately transferred to observation chambers for behavior monitoring, and the number of abdominal constrictions was recorded over the following 30 min. (3) Noxious heat-stimulated tail withdrawal. Mice were orally administered ASA at specified dosages. After 15 min, the mice were gently restrained by the experimenter wearing cotton gloves, and the protruding one-third of the tail was immersed in a water bath maintained at 52 °C. The latency of the tail flick response was recorded. Exposure to the thermal bath was terminated after 10 seconds to prevent tissue damage.

### Statistics & reproducibility

Unless otherwise stated, all the experiments were conducted with at least two independent biological replicates, and consistent results were obtained across these replicates. In practice, the sample size used in this study is usually determined based on the need for it to offer sufficient statistical power, and the time, cost, or convenience of collecting the data. No statistical methods were used to predetermine sample size. No data were excluded from the analyses. The experiments were not randomized. The investigators were not blinded to allocation during experiments and outcome assessment. All samples in this study were allocated randomly. The presentation of data, including sample sizes denoted as biological replicates (n), the conducted statistical analyses, and the significance of differences are shown in the figures, and details are provided in the respective figure legends. For statistical evaluations involving multiple comparisons, GraphPad Prism 8 (v 9.2.0, GraphPad Software Inc.) and Microsoft Excel (v16.51, Microsoft®) were employed. A two-tailed, unpaired Student's t-test and one-way or two-way analysis of variance (ANOVA) were utilized to determine the statistical significance of differences. All the figures were originally created by the authors using Illustrator (V28.2, Adobe, USA). The authors state that no previously created elements were used.

### Reporting summary

Further information on research design is available in the Nature Portfolio Reporting Summary linked to this article.

## Data availability

The authors declare that all the data supporting the findings of this study are available within the paper, supplementary materials and Source Data file. All original plasmids listed in Supplementary Data 1 are available if it is requested for scientific purposes. Any additional inquiries should be addressed to Jinbo Huang or Martin Fussenegger (corresponding author). The full-length sequences of the key plasmids: pJH138, pJH140, pJH142, pJH144, pJH196, pJH230, pJH231, pJH2012, and pJH2036, are available in Supplementary Data 4. Source data are provided with this paper.

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

## Acknowledgements

We are grateful to Pascal Stucheli, Adrian Bertschi and Helene Chassin for project advices, Yu-Qing Xie, Benjamin Danuser for general help, Zhihua Lin for fluorescence microscope operation, Lihang Zhang for animal experiments, Henryk Zulewski for advice on diabetology, and Ghislaine Charpin-El Hamri for assistance with animal experimentation. The work in the laboratory of M.F. is financially supported by a European Research Council (ERC) advanced grant (ElectroGene; grant no. 785800, M.F.) and in part by the Swiss National Science Foundation NCCR Molecular Systems Engineering (M.F.). Work in the laboratory of M.X. is supported by the National Natural Science Foundation of China (NSFC Project 32071429, M.X.) and the Ministry of Science and Technology (MOST Project 2020YFA0909200, M.X.).

## Author contributions

J.H., A.P.T., and M.F. designed the project. J.H. performed all the cell culture experiments. J.H., A.P.T., T.G., S.X., M.X., and M.F. designed the animal experiments. J.H., T.G., and S.X. performed the animal experiments. All authors analysed the results. J.H., A.P.T., M.X and M.F. wrote the manuscript.

## Competing interests

The authors declare no competing interests.
