## [Transparent Peer Review file · Nature Communications]

Aspirin-responsive gene switch regulating therapeutic protein expression

Corresponding Author: Professor Martin Fussenegger

Version 0:

Reviewer comments:

Reviewer #1

(Remarks to the Author)

Summary:

In the Huang et al., manuscript, the authors presented a clever acetylsalicylic acid (ASA)-responsive gene switch utilizing the plant SA receptors NPR1 and NPR4 as the starting materials. Through truncation and mutagenesis of these plant proteins, the authors increased the responsiveness of the switch to ASA in animal cells and proved its application in treating type 1 diabetic mice to decrease diabetic-related marker genes. While the argument the authors made for using their ASPIRIN system for regulating therapeutic protein expression is convincing, several questions remain to be addressed in order to substantiate the claim:

Major concerns:

In line 69, the authors cite that ASA delays Alzheimer's disease. In more recent years, this has been challenged and deemed inconclusive. While it does not directly impact the work the authors have done, they should take caution when stating the uses of ASA. The importance of ASA does not need to be exaggerated to demonstrate the importance of the switch developed.

The author demonstrates that the T2 NPR1 and NPR4 truncations, consisting of mainly the SA binding core (SBC), have the most significant difference in the ASA-induced disruption of their interaction. However, it is not known based on the two cited structural works in plants (i.e., Wang et al., and Kumar et al.) whether NPR1 and NPR4 interact through the SBCs without their known protein-protein interacting BTB and ankyrin repeat domains. The authors should validate the SA- and ASA-mediated association and dissociation of the truncated NPR1 and NPR4 through co-IP and western blotting in addition to their transcriptional assay.

Moreover, the initial work by Fu et al., doi: 10.1038/nature11162 on SA disrupting the NPR1-NPR4 interaction should be cited.

Multiple figures are missing statistics or proper statistics. This makes some of the statements made by the authors impossible to validate. The authors claim that the mutants developed are more responsive to SA or ASA (Figure 2b, d, f, and h). However, without a two-way ANOVA comparing the mutant and the wild type, the authors cannot make the claim.

In Figure 2i-k, the condition of the cells used in the negative control appears very different from the SA and ASA-treated cells, where the negative control cell is larger and does not have other cells around it. Do the authors have images showing these cells under more similar conditions? Furthermore, the authors should quantify the images of multiple cells and perform statistics to demonstrate the differences in protein colocalization.

In lines 179-183, the authors state the EC50 SA concentration of the system is lower than the blood concentration. Is the EC50 in cell culture versus the blood concentrations in animals a fair comparison?

The authors stated that many of the diabetic neuropathy behaviors are alleviated by ASA, but did not measure these behaviors using the diabetic mice, making it difficult to compare the results Figure 7a-c with Figure 7d-i. Are the effects observed in Figure 7d-i due to the ASA treatment or ASA induced ASPIRIN activation or both? It seems that T1D NC with ASA treatment should be included as a control to distinguish the effect of ASA from that of ASA-induced insulin production.

The authors state that the mutations that occur are in the SA binding pocket. As negative controls, why did the authors not use a mutated NPR4 with the inability to perceive SA? The authors also state that the cells do not show abnormalities over 1 nM. But many of the concentrations used are higher than this.

Minor concerns:

Figure 1a seems unneeded given the authors did not discover the structural similarity between SA and ASA.

In lines 124-126, the authors claim to establish NPR1 and NPR4 as homologs with SBC. This has been known for several years and was not established by the authors. The authors should change the wording from established to "As previously shown" or something similar to demonstrate the authors are not the first to establish this.

Figure 5b: Since the authors did not discover this pathway, references need to be cited or the figure panel should be moved to supplemental information.

In line 165, ASA disrupts this interaction, so "meditating" is not the best word choice.

Include the p-values between T1D NC and T1D ASA for Figures 7d-g even if they are not significant.

The abbreviation i.p. is used twice for two different terms (lines 288 [intraperitoneally] and 294 [implanted])

T1D is used before defining type 1 diabetes.

In plants, INA and BTH also mimic SA. Do INA and BTH function similarly in this system?

For line 228, please replace nearly 100% with the actual percentage.

Reviewer #2

(Remarks to the Author)

Summary:

In this manuscript, the authors introduce an acetylsalicylic acid (ASA or aspirin) sensor using NPR1 and NPR4 fused with a DNA binding domain (VanR) and a transactivation domain (VP16). This new sensor was demonstrated to function in vivo to regulate insulin expression and to provide pain relief in animals. Overall, this is an exciting new tool that has the potential to be further developed.

Comments for authors:

1. The introduction of the mechanisms of SA, ASA, and COX enzymes is confusing. I suggest making a schematic and either incorporate it in the first figure or offer it in the supplemental information.
2. More details on the controls in experiments are needed. For example, in Figure 1, there is reference to the control sample, but it is not clear what constitutes the control.
3. There is a mention of the acetic acid-evoked withering test. What is this? One or two sentences explaining what this test is will help the reader to understand what this is.
4. I'm curious about the variation of background and potential leakage of the system. Can the authors comment on this? Leakage would be an important characteristic to discuss, especially if this is to be developed into a therapeutic.

Reviewer #3

(Remarks to the Author)

Gene- and cell-based therapies are a hot topic and a compelling need for a range of conditions. This paper shows great promising results of a new ASA-induced gene switch system on the induction and fine tuning of gene expression. Together with the reporter gene, author test the expression of insulin in an in vivo model of T1D, obtaining improving of glycemic control and diabetic complications. All results are consistent and clearly presented and support the conclusions. In particular, the optimization of the ASA-induced system denotes a remarkable and complicated amount of work and pends-up with a great sensitive, reversible and tunable transgene platform. Data from this paper could pave the way to develop new cell-based gene therapies, helping the management of different disease. Below, just some flaws to be addressed before publication.

Results:

- Fig. Suppl. 8 b-e: add percentage to the dot plot quadrants.
- Fig. Suppl. 8 and 9: legend refers to n=4 samples, but in panels 8f, 9f and 9k there are just three dots.
- Results section, paragraph 4: Describing information about the selected clone and its usage for the following experiments also in this section and not only in the legend of Fig. Suppl. 9, would be helpful for the reader.
- For experiments in Fig.6 ASA was administered i.p., while for experiments in Fig. 7a-c the drug was given orally. Moreover, ASA concentration and way of administration were missing for experiments in Fig. 7 d-l. Please, add these data.
- Fig.7: Data of biomarkers levels from not-implanted T1D mice treated with ASA are missing. Insulin is known to have some anti-inflammatory properties, it would be nice to clarify if anti-inflammatory effects shown in T1D transplanted animals treated

with ASA would be of different extent in T1D not-implanted animals treated with the drug.

Discussion:

- Authors hypothesize the possible use of their ASPIRIN-system for different conditions.

Could you indicate ASA human dosage that would correspond to ASA concentration used to have significant results in in vivo experiments? Would this dosage have some side effects in human if used for long periods?

- References on diabetic complications do not include paper on cardiovascular complication and diabetic retinopathy.

Authors could refer to the following papers for a more complete overview on all the complications: Pastore, I. et al, The Impact of Diabetes Mellitus on Cardiovascular Risk Onset in Children and Adolescents. *Int. J. Mol. Sci.* 2020, 21, 4928. <https://doi.org/10.3390/ijms21144928>; D'Addio, F., et al. Abnormalities of the oculomotor function in type 1 diabetes and diabetic neuropathy. *Acta diabetologica* (2022)., 59(9), 1157–1167. <https://doi.org/10.1007/s00592-022-01911-1>).

M&M:

- Specify the strain of the mice

Reviewer #4

(Remarks to the Author)

The current study proposes a novel approach by employing aspirin-based gene therapy to improve diabetes and its complications. However, there are some concerns to be addressed:

The current study showed the effects of the HEK-ASPIRIN implantation only for four weeks. In order to demonstrate its potential clinical relevance, it would be desired to confirm the effect for a longer period. Is the glucose-lowering effect sustained for a longer period by the single implantation or the repeated ones?

In lines 302-304 of the manuscript, the authors stated that the implantation reduced postprandial blood glucose elevation, which implies that the implanted cells showed increased insulin secretion in response to glucose. The authors should provide the functional characteristics of the implanted cells regarding glucose-stimulated insulin secretion using the glucose challenge test (ipGTT or OGTT).

While the authors have indicated some serum proteins related to diabetic neuropathy, they may not be sufficient to demonstrate therapeutic efficacy for the prevention of diabetic complications. If possible, it would be nice to perform physiological evaluations, such as nerve conduction studies or pathological examinations.

Version 1:

Reviewer comments:

Reviewer #1

(Remarks to the Author)

The authors have addressed all my concerns satisfactorily. I would like to see this exciting work in press soon.

Reviewer #2

(Remarks to the Author)

The authors have provided adequate responses to my comments and requests for clarification in the manuscript. The manuscript is much clearer.

Revision Responses for NCOMMS-24-41573-T

Point-by-point responses to referees.

Reviewer #1 (Remarks to the Author):

Summary:

In the Huang et al., manuscript, the authors presented a clever acetylsalicylic acid (ASA)-responsive gene switch utilizing the plant SA receptors NPR1 and NPR4 as the starting materials. Through truncation and mutagenesis of these plant proteins, the authors increased the responsiveness of the switch to ASA in animal cells and proved its application in treating type 1 diabetic mice to decrease diabetic-related marker genes. While the argument the authors made for using their ASPIRIN system for regulating therapeutic protein expression is convincing, several questions remain to be addressed in order to substantiate the claim:

Thank you for your positive feedback on our work and for recognizing the innovative aspects of our ASPIRIN-responsive gene switch based on plant SA receptors.

Major concerns:

1. In line 69, the authors cite that ASA delays Alzheimer's disease. In more recent years, this has been challenged and deemed inconclusive. While it does not directly impact the work the authors have done, they should take caution when stating the uses of ASA. The importance of ASA does not need to be exaggerated to demonstrate the importance of the switch developed.

Thanks for pointing this out. We have removed the statement.

2a. The author demonstrates that the T2 NPR1 and NPR4 truncations, consisting of mainly the SA binding core (SBC), have the most significant difference in the ASA-induced disruption of their interaction. However, it is not known based on the two cited structural works in plants (i.e., Wang et al., and Kumar et al.) whether NPR1 and NPR4 interact through the SBCs without their known protein-protein interacting BTB and ankyrin repeat domains. The authors should validate the SA- and ASA-mediated association and dissociation of the truncated NPR1 and NPR4 through co-IP and western blotting in addition to their transcriptional assay.

We agree that demonstrating direct protein-protein interaction between the truncated versions of NPR1 and NPR4 would provide stronger evidence for the mechanistic interpretation of our findings. Therefore, in addition to our transcriptional assay, we newly performed co-IP and western blotting analyses. These results further confirmed the SA- and ASA-mediated association and dissociation of the truncated NPR1 and NPR4, and we have included the data in **Supplementary Figure 8** in the revised manuscript.

2b. Moreover, the initial work by Fu et al., doi: 10.1038/nature11162 on SA disrupting the NPR1-NPR4 interaction should be cited.

Thank you. We have included this reference.

3. Multiple figures are missing statistics or proper statistics. This makes some of the statements made by the authors impossible to validate. The authors claim that the mutants developed are more responsive to SA or ASA (Figure 2b, d, f, and h). However, without a two-way ANOVA comparing the mutant and the wild type, the authors cannot make the claim.

Thanks for the valuable feedback regarding the statistical analysis. We have now performed two-way ANOVA to compare the mutants and wild type across treatments. We have also revised the description of the results to improve the clarity. The updated statistical results, including *p*-values, have been incorporated in the revised manuscript (new **Figure 2**).

4. In Figure 2i-k, the condition of the cells used in the negative control appears very different from the SA and ASA-treated cells, where the negative control cell is larger and does not have other cells around it. Do the authors have images showing these cells under more similar conditions? Furthermore, the authors should quantify the images of multiple cells and perform statistics to demonstrate the differences in protein colocalization.

Thank you for your observation. In response, we have updated **Figure 2** with new microscopy images having matched cell densities and included the conditions for the negative control.

Additionally, we have quantified protein colocalization across multiple cells and performed statistical analysis to confirm the differences. These quantifications and statistical results, along with representative images, have been incorporated into new **Figure 2i-n** in the revised manuscript.

5. In lines 179-183, the authors state the EC₅₀ SA concentration of the system is lower than the blood concentration. Is the EC₅₀ in cell culture versus the blood concentrations in animals a fair comparison?

Thanks for bringing this to our attention. We agree that comparing the EC₅₀ derived from an in vitro system to in vivo blood concentrations is problematic, as the in vitro conditions do not fully replicate the complexity of the in vivo environment, including factors such as tissue distribution, metabolism, and protein binding, which can influence the effective concentration of SA and ASA in the blood. To avoid potential misunderstanding, we have removed this comparison in our revised manuscript.

6. a. The authors stated that many of the diabetic neuropathy behaviors are alleviated by ASA, but did not measure these behaviors using the diabetic mice, making it difficult to compare the results Figure 7a-c with Figure 7d-l. Are the effects observed in Figure 7d-l due to the ASA treatment or ASA induced ASPIRIN activation or both? b. It seems that T1D NC with ASA

treatment should be included as a control to distinguish the effect of ASA from that of ASA-induced insulin production.

Thank you for your insightful feedback.

a. We acknowledge the absence of direct behavioral measurements in the diabetic mice in Figure 7d-i. However, our intention was not to directly compare the results of Figure 7a-c (new **Supplementary Figure 16**) with those of Figure 7d-i (new **Figure 7a-i**). Rather, the purpose of Figure 7a-c was to confirm the well-established anti-nociceptive activity of ASA, as documented in previous studies (Al - Swayeh O. et al., *British Journal of Pharmacology* 2000; Mohamad A. et al., *European Journal of Pharmacology* 2010; Spindola et al. *BMC Pharmacology* 2010; Z.A. Zakaria et al. *Revista Brasileira de Farmacognosia* 2016; Hui M.O. et al., *Scientific Reports* 2021). We combined the behavioral measurements in wild-type mice (original Figure 7 a-c) and the ASPIRIN performance data in the diabetic mice (original Figure 7 d-i) in the same figure to emphasize that the inducer used in our system has dual functions: anti-nociceptive activity for pain management and therapeutic protein-inducing activity for gene therapy. However, taking into account your minor points 1 and 3 as well, and since the findings in original Figures 7a-c are not novel, we have moved the original **Figure 7a-c** to the supplementary data (new **Supplementary Figure 16**) in the revised manuscript to avoid potential misunderstanding.

In our study, type 1 diabetic mice with sustained high blood glucose for over three months exhibited elevated levels of biomarkers associated with diabetic neuropathy. Our system, which uses aspirin as an insulin inducer to reduce blood glucose (**Figure 6**), not only restored normoglycemia but also improved diabetic neuropathy-related biomarkers (new **Figure 7**). We have revised the manuscript to clarify that aspirin alleviated diabetic neuropathy-related biomarkers, making no claim regarding neuropathic behaviors. It is important to bear in mind that, physiologically, diabetic neuropathy is typically considered irreversible, as nerve damage caused by prolonged high blood glucose does not fully regenerate. This has been extensively studied, for example by Sloan et al., *Nature Reviews Endocrinology*, 2021; Said, *Nature Clinical Practice Neurology*, 2007; Vinik, *The American Journal of Medicine*, 1999; Said, *Handbook of Clinical Neurology*, 2013.

b. In response to your suggestion, we agree that a T1D NC (negative control) group with ASA treatment as a control is needed to distinguish the effects of ASA alone versus ASA-induced insulin production. Reviewer 3 also had a similar concern (please also see our response to point 5 of Reviewer 3). We have now included the data from a T1D NC group in our revised **Figure 7**. We have also revised the text accordingly.

7. The authors state that the mutations that occur are in the SA binding pocket. **a.** As negative controls, why did the authors not use a mutated NPR4 with the inability to perceive SA? **b.** The authors also state that the cells do not show abnormalities over 1 nM. But many of the concentrations used are higher than this.

We appreciate the reviewer's insightful comments.

a). Use of mutated NPR4 as negative control: To address this point, we designed point mutations within the SA-binding pocket of NPR4. Our results confirmed that the mutated NPR4 variants had completely lost the ability to bind both ASA and SA. These findings have been included in the revised manuscript as **Supplementary Figure 6**. We agree that using the mutated NPR4 as a negative control is logically sound. However, due to the higher basal leakiness of mutated NPR4 compared to the evolved variant (as shown in **Supplementary Figure 6**), we opted not to use it in this study in order to maintain experimental consistency and reliability.

b). Clarification on concentration ranges: After reviewing our original manuscript and data, we confirmed that the cells showed no abnormalities when exposed to concentration up to 250 μM ASA or SA, as presented in original Supplementary Figure 2 f-i (now **Supplementary Figure 3 f-i**) and Supplementary Figure 6 a-d (now **Supplementary Figure 9 a-d**), based on cell viability assessment. Moreover, ASA or SA concentrations ranging from 50 to 250 μM had no significant impact on either the rate of cell growth, or the capability for constitutive expression of NLuc, as shown in original Supplementary Figure 7 a-d (now **Supplementary Figure 10a-d**). We have corrected the revised manuscript.

Minor concerns:

1. Figure 1a seems unneeded given the authors did not discover the structural similarity between SA and ASA.

We agree, and have removed the structures from **Figure 1a** in the revised manuscript. However, for the convenience of readers, we have included smaller representations of these structures within the schematic models (new **Figure 1b**), in the same way that the molecular structures are presented in **Figure 5a**.

2. In lines 124-126, the authors claim to establish NPR1 and NPR4 as homologs with SBC. This has been known for several years and was not established by the authors. The authors should change the wording from established to “As previously shown” or something similar to demonstrate the authors are not the first to establish this.

Thank you. We have clarified this.

3. Figure 5b: Since the authors did not discover this pathway, references need to be cited or the figure panel should be moved to supplemental information.

Thank you. We have added appropriate citations to the figure legend, and moved this panel to the supplemental information.

4. In line 165, ASA disrupts this interaction, so “mediating” is not the best word choice.

We have now used the term “inhibited”.

5. Include the p-values between T1D NC and T1D ASA for Figures 7d-g even if they are not significant.

Done.

6. The abbreviation i.p. is used twice for two different terms (lines 288 [intraperitoneally] and 294 [implanted]). T1D is used before defining type 1 diabetes.

We have corrected this issue.

7. In plants, INA and BTH also mimic SA. Do INA and BTH function similarly in this system?

In line with our tests of ASA analogues presented in Figure 5a, we conducted in vitro cell experiments using INA (2,6-dichloroisonicotinic acid) and BTH (benzothiadiazole) as inducers in our monoclonal cell lines to assess their functionality in our system. The results indicate that neither INA nor BTH has inducing activity in our experimental setup. We have included these data in the revised **Figure 5a** as well as in new **Supplementary Figure 14a**.

8. For line 228, please replace nearly 100% with the actual percentage.

Done.

Reviewer #2 (Remarks to the Author):

Summary:

In this manuscript, the authors introduce an acetylsalicylic acid (ASA or aspirin) sensor using NPR1 and NPR4 fused with a DNA binding domain (VanR) and a transactivation domain (VP16). This new sensor was demonstrated to function *in vivo* to regulate insulin expression and to provide pain relief in animals. Overall, this is an exciting new tool that has the potential to be further developed.

Thank you for your thoughtful consideration of our manuscript. We are most grateful for your helpful comments.

Comments for authors:

1. The introduction of the mechanisms of SA, ASA, and COX enzymes is confusing. I suggest making a schematic and either incorporate it in the first figure or offer it in the supplemental information.

Thank you for your suggestion. We have included a schematic representation to show the relevant mechanisms in the new **Supplementary Figure 1** in our revised manuscript.

2. More details on the controls in experiments are needed. For example, in Figure 1, there is reference to the control sample, but it is not clear what constitutes the control.

To clarify this, we have replaced “control” with “non-induced” for consistency, and revised the figure legends accordingly (new **Figure 1e-h**, **Figure 2i**, **Figure 3b**, **Figure 4b**, and new **Figure S3b-I**, **Figure S4a-d**, **Figure S5a-d**, **Figure S6**, **Figure S7a,d**, **Figure S9c**, **Figure S11g**, and **Figure S12a**).

3. There is a mention of the acetic acid-evoked writhing test. What is this? One or two sentences explaining what this test is will help the reader to understand what this is.

The acetic acid-evoked writhing test is a commonly used animal model for evaluating analgesic (pain-relieving) activity (Al - Swayeh O. et al., *British Journal of Pharmacology* 2000; Mohamad A. et al. *European Journal of Pharmacology* 2010.). In this test, an intraperitoneal injection of acetic acid induces a characteristic writhing response in mice, which serves as an indicator of visceral pain. The frequency of these writhes is measured to assess the effectiveness of analgesic treatments. We have included a concise explanation in our manuscript.

4. I'm curious about the variation of background and potential leakage of the system. Can the authors comment on this? Leakage would be an important characteristic to discuss, especially if this is to be developed into a therapeutics.

Thank you for your input.

Background variation: Background variations could arise from multiple factors, including plasmid ratios, cell type and conditions, transfection efficiency, differences in expression time,

reporter substrate stability, and unpredictable technical variables (Esposito, M. et al. *Eur J Oral Sci.*, 1998; Hilfinger A. and Johan P., *Proceedings of the National Academy of Sciences*; Iwasaki W.M., et al., *BMC Ecology and Evolution* 2013; Woodward J., *Biology & Philosophy* 2010). We have implemented rigorous controls and standardized protocols throughout our experiments to minimize these variations. Furthermore, we have provided detailed descriptions of the methodologies used to ensure reproducibility.

Potential leakage: Leakage is indeed a critical consideration for effectiveness and safety. We have conducted various preliminary assessments aimed at minimizing leakage in our system. For instance, we performed several rounds of mutagenesis to optimize the system, which not only increased expression levels, but also decreased baseline expression levels. As a result, the level of leakage in our system is now negligible, and is comparable to, or even better than, the levels in other similar studies (Zhou Y et al., *Nature Biotechnology* 2021; Krawczyk K. et al., *Science*, 2020; Chen H et al., *Nature Biotechnology* 2024a; Chen H et al., *Nature Biotechnology* 2024b).

We have expanded our discussion regarding leakage in the revised manuscript to provide greater clarity on this important issue.

Reviewer #3 (Remarks to the Author):

Gene- and cell-based therapies are a hot topic and a compelling need for a range of conditions. This paper shows great promising results of a new ASA-induce gene switch system on the induction and fine tuning of gene expression. Together with the reporter gene, author test the expression of insulin in an in vivo model of T1D, obtaining improving of glycemic control and diabetic complications. All results are consistent and clearly presented and support the conclusions. In particular, the optimization of the ASA-induced system denotes a remarkable and complicated amount of work and pends-up with a great sensitive, reversible and tunable transgene platform. Data from this paper could pave the way to develop new cell-based gene therapies, helping the management of different disease. Below, just some flaws to be addressed before publication.

Thank you very much for your encouraging and positive feedback on our study. We appreciate your helpful comments.

Results:

1. Fig. Suppl. 8 b-e: add percentage to the dot plot quadrants.

Done.

2. Fig. Suppl. 8 and 9: legend refers to n=4 samples, but in panels 8f, 9f and 9k there are just three dots.

Thanks for pointing this out. We have revised the figure legends accordingly.

3. Results section, paragraph 4: Describing information about the selected clone and its usage for the following experiments also in this section and not only in the legend of Fig. Suppl. 9, would be helpful for the reader.

Thank you. As you suggested, we have incorporated this information directly into the Results section, specifically in paragraph 4. We agree this addition provides readers with a better understanding of the clone's significance and its role in our experimental design, rather than relying solely on the figure legend of Supplementary Figure 9 (new **Supplementary Figure 12**).

4. For experiments in Fig.6 ASA was administered i.p., while for experiments in Fig. 7a-c the drug was given orally. Moreover, ASA concentration and way of administration were missing for experiments in Fig. 7 d-l. Please, add these data.

Thanks for noticing this. We have clarified the routes of administration and ASA concentrations in the revised manuscript. For the experiments in **Figure 6**, ASA was administered intraperitoneally (i.p.), as previously mentioned. For **Figure 7a-c** (now **Supplementary Figure 16**), the drug was given orally, by following the standard protocols used in previous

studies (Al - Swayeh O. et al., *British Journal of Pharmacology* 2000; Mohamad A. et al., *European Journal of Pharmacology* 2010; Spindola et al. *BMC Pharmacology* 2010; Z.A. Zakaria et al. *Revista Brasileira de Farmacognosia* 2016; Hui M.O. et al., *Scientific Reports* 2021) (please also see the response to major point 6 of Reviewer 1).

We have also added the missing details for **Figure 7d-l** (new **Figure 7a-i**).

5. Fig.7: Data of biomarkers levels from not-implanted T1D mice treated with ASA are missing. Insulin is known to have some anti-inflammatory properties, it would be nice to clarify if anti-inflammatory effects shown in T1D transplanted animals treated with ASA would be of different extent in T1D not-implanted animals treated with the drug.

Thank you for your insightful feedback, which aligns with point 6 raised by Reviewer 1. We have now incorporated the relevant data into the revised **Figure 7**. The effects observed in Figure 7d-l (new **Figure 7a-i**) likely result from the combination of ASA treatment itself and ASA-induced insulin production, as anticipated. We have also revised the text to explain this more clearly (please also see the major point 6 of Reviewer 1).

6. Discussion:

6.1 Authors hypothesize the possible use of their ASPIRIN-system for different conditions. Could you indicate ASA human dosage that would correspond to ASA concentration used to have significant results in in vivo experiments? Would this dosage have some side effects in human if used for long periods?

Thank you for your thoughtful questions. To estimate a comparable human equivalent dosage (HED), we utilized standard dose conversion based on body surface area (BSA), employing a conversion factor derived from the relative surface area of the species, as documented in clinical studies (Reagan-Shaw et al., *FASEB J.* 2008; Nair A.B. and Jacob S., *J Basic Clin Pharm.* 2016). The conversion formula is:

$$\text{HED (mg/kg)} = \frac{\text{Animal dose (mg/kg)} \times \text{Mouse Km factor (3)}}{\text{Human Km factor (37)}}$$

In our experiments, the effective ASA dosage in mice is approximately 10-100 mg/kg. This translates to an estimated human equivalent dose of 48.6-486 mg/day for an average adult weighing 60 kg. This calculated converted dosage coincides well with the therapeutic range for aspirin in humans, commonly 75-325 mg/day for cardiovascular protection (Patrono C., *New England Journal of Medicine* 1994; Campbell C.L. et al., *JAMA* 2007), and is significantly lower than the dosages of 1200-2600 mg/day typically used for treating mild to moderate pain, such as headaches and muscle aches (Nalamachu S., *Am J Manag Care* 2013; Amaechi O. et al., *American Family Physician* 2021; Searle S. et al., *The Clinical Journal of Pain* 2020).

Long-term administration of aspirin at higher doses may pose risks, including gastrointestinal irritation, bleeding, and renal impairment (Harirforoosh S. et al., *Journal of Pharmacy & Pharmaceutical Sciences* 2013; Vonkeman H.E. and van de Laar M.A., *Seminars in Arthritis*

and Rheumatism 2010). However, the estimated dose for therapeutic effects in our system remains within clinically established ranges.

We have included this discussion and the estimated HED in our revised manuscript to clarify the translational potential and safety considerations of our system.

6.2 References on diabetic complications do not include paper on cardiovascular complication and diabetic retinopathy. Authors could refer to the following papers for a more complete overview on all the complications: Pastore, I. et al, The Impact of Diabetes Mellitus on Cardiovascular Risk Onset in Children and Adolescents. *Int. J. Mol. Sci.* 2020, 21, 4928. <https://doi.org/10.3390/ijms21144928>; D'Addio, F., et al. Abnormalities of the oculomotor function in type 1 diabetes and diabetic neuropathy. *Acta diabetologica* (2022)., 59(9), 1157–1167. <https://doi.org/10.1007/s00592-022-01911-1>).

Thank you. We have cited the references you kindly suggested.

7. M&M: - Specify the strain of the mice.

Done.

Reviewer #4 (Remarks to the Author):

The current study proposes a novel approach by employing aspirin-based gene therapy to improve diabetes and its complications.

Thank you. We appreciate your helpful comments on our study.

However, there are some concerns to be addressed:

1. The current study showed the effects of the HEK-ASPIRIN implantation only for four weeks. In order to demonstrate its potential clinical relevance, it would be desired to confirm the effect for a longer period. Is the glucose-lowering effect sustained for a longer period by the single implantation or the repeated ones?

We wish to emphasize that our in vivo experiment was designed as a proof-of-concept study to validate the dual functionality of the HEK-ASPIRIN system in mice, not as a demonstration of clinical relevance. In this context, we believe the four-week duration provides sufficient evidence to demonstrate its effectiveness in regulating glucose levels and to support our conclusions. Many studies in the field have adopted a similar approach, some with even shorter experimental durations, including for example, but not limited to: Kemmwer C. et al., *Nature Biotechnology* 2010, 7 days; Ye H. et al., *Science* 2011, 3 days; Rössger K. et al., *Nature Communications*, 2013, 5 days; Xie M. et al., *Science* 2016, 21 days; Liu Y. et al., *Cell* 2018, 3 days; Bai P. et al., *Nature Medicine* 2019, 35 days; Krawczyk K. et al., *Science* 2020, 7 days; Mansouri M. et al., *Nature Communications* 2021, 12 days; Chen C. et al., *Nature Chemical Biology* 2022, 30 days; Huang J. et al., *Nature Metabolism* 2023, 35 days; Zhao H. et al., *Lancet Diabetes Endocrinol.* 2023, 7 days; Wang X. et al., *Nature Chemical Biology* 2023, <3 days; Ma X. et al., *Molecular Cell* 2024, 30 days.

Furthermore, extending the study would raise ethical concerns regarding animal welfare, since the primary objectives of the experiment—validation of glucose control and management of pain-related biomarkers—have already been successfully demonstrated within the four-week period. We are required to comply with Swiss Federal Regulations (which are identical to European Union Directive EU2019/1010) enforcing the "Three Rs" principle - Replacement, Reduction, and Refinement - to minimize animal use and suffering in scientific procedures.

We feel that the data that we have provided robustly demonstrate consistent glucose-lowering effects over the four-week period following a single implantation, serving to validate the functionality of the switch.

2. In lines 302-304 of the manuscript, the authors stated that the implantation reduced postprandial blood glucose elevation, which implies that the implanted cells showed increased insulin secretion in response to glucose. The authors should provide the functional characteristics of the implanted cells regarding glucose-stimulated insulin secretion using the glucose challenge test (ipGTT or OGTT).

Thank you. We agree that profiling the functional characteristics of the implanted cells using glucose tolerance tests (GTT) is important. In fact, we had already conducted ipGTT to assess postprandial blood glucose regulation and the data were included in the original manuscript (see **Figure 6f**). This figure clearly demonstrates the glucose-lowering effect and the functional insulin response following glucose challenge in the implanted group.

3. While the authors have indicated some serum proteins related to diabetic neuropathy, they may not be sufficient to demonstrate therapeutic efficacy for the prevention of diabetic complications. If possible, it would be nice to perform physiological evaluations, such as nerve conduction studies or pathological examinations.

Thank you for this comment. We agree that the management of relevant biomarkers alone is insufficient to establish therapeutic efficacy in preventing diabetic complications such as neuropathy. However, we should like to emphasize that the primary goal of this proof-of-concept study was not to demonstrate therapeutic efficacy for prevention of neuropathy. Rather, the aims of the in vivo experiment were as follows. i). To demonstrate the immediate therapeutic benefits of the aspirin-controlled gene switch—specifically to highlight the reversal of hyperglycemia, which is the root cause of diabetic neuropathy, and the control of biomarkers associated with diabetic complications, using type 1 diabetes as a complex model. ii). To highlight aspirin’s dual role—both as a trigger for insulin expression and as a painkiller for neuropathy-associated pain. Aspirin’s analgesic properties are well-documented (Jack D. B. *The Lancet* 1997; Schmidt M. et al. *European Heart Journal* 2016; Schjerning A. M. et al., *Nature Reviews Cardiology* 2020; Spiegel R. *Nature Reviews Cardiology* 2020).

The key outcome of this experiment is that the HEK-ASPIRIN system successfully restored normoglycemia and decreased relevant biomarkers in mice that had experienced prolonged hyperglycemia for over three months and that showed elevated biomarkers linked to diabetic neuropathy (see Iyengar, S., *Pain* 2017; Navarro-Gonzalez, J. F. and Mora-Fernandez, C., *Journal of the American Society of Nephrology* 2008; Bhutia, Y. et al., *Journal of Natural Science, Biology, and Medicine* 2011; Goldberg, R. B., *The Journal of Clinical Endocrinology & Metabolism* 2009; Lontchi-Yimagou, E. et al., *Current Diabetes Reports* 2013). The connection between hyperglycemia and diabetic neuropathy is well-established (G. Sloan et al., *Nature Reviews Endocrinology* 2021; G. Said, *Nature Clinical Practice Neurology* 2007; A. I. Vinik, *The American Journal of Medicine* 1999; G. Said, *Handbook of Clinical Neurology* 2013).

We appreciate your suggestion, and certainly we agree that additional physiological evaluations will be needed to evaluate the therapeutic efficacy of our system during potential clinical translation, but we feel this is beyond the scope of the present work. To avoid misunderstanding, we have revised the manuscript to clarify that our system decreases biomarkers related to diabetic neuropathy without claiming that it can necessarily prevent or reverse nerve damage (please also refer to point 6a of Reviewer 1).